

# Aerosol-Cloud Closure Study on Cloud Optical Properties using Remotely Piloted Aircraft Measurements during a BACCHUS Field Campaign in Cyprus

Radiance Calmer[1], Gregory C. Roberts[1,2], Kevin J. Sanchez[1,2], Jean Sciare[3], Karine Sellegri[4], David Picard[4], Mihalis Vrekoussis[3,5,6], and Michael Pikridas[3]

[1]Centre National de Recherches Météorologiques (CNRM), UMR 3589, Météo-France/CNRS, Toulouse, France
[2]Scripps Institution of Oceanography, University of California, San Diego, CA;
[3]Energy, Environment and Water Research Center, The Cyprus Institute, 2121 Nicosia, Cyprus
[4]LaMP, Laboratoire de Météorologie Physique CNRS UMR6016, Observatoire de Physique du Globe de Clermont-Ferrand, Université Clermont Auvergne, Aubière, France
[5]Institute of Environmental Physics and Remote Sensing (IUP-UB), University of Bremen, 28359 Bremen, Germany
[6]Center of Marine Environmental Sciences (MARUM), University of Bremen, 28359 Bremen, Germany

**Correspondence:** Gregory C. Roberts (greg.roberts@meteo.fr)

**Abstract.** In the framework of the EU-FP7 BACCHUS project, an intensive field campaign was performed in Cyprus (2015/03). Remotely Piloted Aircraft System (RPAS), ground-based instruments, and remote-sensing observations were operating in parallel to provide an integrated characterization of aerosol-cloud interactions. Remotely Piloted Aircraft (RPA) were equipped with a 5-hole probe, pyranometers, pressure, temperature and humidity sensors, and measured updraft velocity at cloud base

and cloud optical properties of a stratocumulus layer. Ground-based measurements of dry aerosol size distributions and cloud condensation nuclei spectra, and RPA observations of vertical wind velocity and meteorological state parameters are used here to initialize an Aerosol-Cloud Parcel Model (ACPM) and compare the in situ observations of cloud optical properties measured by the RPA to those simulated in the ACPM. Two different cases are studied with the ACPM, including an adiabatic case and an entrainment case, in which the in-cloud temperature profile from RPA is taken into account. Adiabatic ACPM simulation yields

cloud droplet number concentrations at cloud base (ca. 400 cm$^{-3}$) that are similar to those derived from a Hoppel minimum analysis. Cloud optical properties have been inferred using the transmitted fraction of shortwave radiation profile measured by downwelling and upwelling pyranometers mounted on a RPA, and the observed transmitted fraction of solar radiation is then compared to simulations from the ACPM. ACPM simulations and RPA observations show better agreement when associated with entrainment compared to that of an adiabatic case. The mean difference between observed and adiabatic profiles

of transmitted fraction of solar radiation is 0.12, while this difference is only 0.03 between observed and entrainment profiles. A sensitivity calculation is then conducted to quantify the relative impacts of two-fold changes in aerosol concentration, and updraft velocity to highlight the importance of accounting for the impact of entrainment in deriving cloud optical properties, as well as the ability of RPAs to leverage ground-based observations for studying aerosol-cloud interactions.





# 1 Introduction

The influence of aerosol-cloud interactions on the climate is through the first indirect aerosol effect (Twomey, 1974), the second indirect aerosol effect (Albrecht, 1989), and other effects of aerosols on cloud (a comprehensive review is given in Lohmann and Feichter (2004)). As the Intergovernmental Panel on Climate Change (IPCC) report (Boucher et al., 2013) aims to quantify

the effective radiative forcing due to aerosol-cloud interactions, discrepancies still remain between observations and model results. Even though the Twomey or cloud-albedo effect might be considered as the most studied, discussions are still on-going to better understand the correlation between cloud condensation nuclei (CCN), supersaturation ($S$), vertical wind velocity ($w$), cloud droplet number concentration ($N_d$), and the impact on cloud albedo depending on the environmental conditions (Hudson and Noble (2014a); Werner et al. (2014); Cecchini et al. (2017); Sarangi et al. (2018)). With the support of fifteen years of

satellite measurements, calculation of albedo susceptibilities helps to better understand the cloud radiative response due to aerosol-cloud interactions, and supports the conclusion that polluted clouds less efficiently change their albedo compared to more pristine clouds for the same change in $N_d$ (Painemal, 2018). Cloud droplet number concentrations have been the center of interest for satellite retrieval calculations based on cloud optical depth, cloud droplet effective radius, and cloud top temperature. Nonetheless, a high relative uncertainty is still associated with $N_d$ (Grosvenor et al., 2018). Bender et al. (2016) also showed

disagreement between model and satellite observations for the influence of aerosol loading on cloud albedo. Consequently climate models tend to overestimate the albedo compared to observations when the contribution of aerosol was considered. Ma et al. (2018) identified steps in satellite retrieval procedures, which led to errors in cloud susceptibilities to aerosols, and biased comparison with climate model. More generally, a call for more validation studies in different cloud regimes with in situ data has been expressed (Grosvenor et al., 2018), specifically to provide the whole cloud profile and detailed picture of the causes

of differences between in situ measurements, satellite retrievals, and model simulations.

Traditionally, manned aircraft have been used to conduct aerosol-cloud closure studies, where a closure experiment aims to characterize the same parameters of a system with different, independent methods and models to minimize the measurement uncertainties through comparison of derived values (i.e., Weinzierl et al. (2017)). Closure studies mainly focus on comparisons

between cloud droplet number concentration, obtained from in situ measurements, and calculated from an Aerosol-Cloud Parcel Model (ACPM). Conant et al. (2004) presented the first study to achieve a closure within 15 % for cumulus clouds of marine and continental origin during the CRYSTAL-FACE experiment (Key West, Florida, July 2002). Meskhidze et al. (2005) also obtained a good agreement, within 30 %, for stratocumulus clouds (CSTRIPE, Monterey, California, July 2003). For a highly polluted environment (ICARTT, Detroit, Michigan, Cleveland, Ohio, 2004), Fountoukis et al. (2007) achieved a

closure within 10 % on average. These studies also highlight that cloud droplet number concentrations are more sensitive to aerosol and upraft velocity depending on atmospheric conditions. The aerosol- and updraft-limited regimes for cloud droplet formation were studied with an adiabatic parcel model in Reutter et al. (2009), and a differentiation between the regimes was proposed based on the relative sensitivity ratios ($dlnN_d/dlnw$ and $dlnN_d/dlnN$). The results of the model were consistent with field observations in clean/polluted environments (Fountoukis et al. (2007); Hudson and Noble (2014a); Hudson and No-



ble (2014b)). Previous closure studies on $N_d$ were only conducted with adiabatic simulations, even if already pointed out in Conant et al. (2004) that 1) the effects of entrainment mixing had to be included for a more comprehensive description of cloud microphysics, and 2) nearly adiabatic profiles were maintained only through the lowest part of the cloud. To address some of the discrepancies in previous studies, a BACCHUS (impact of Biogenic versus Anthropogenic emissions on Clouds and Climate: towards a Holistic UnderStanding) field campaign took place at Mace Head, Ireland, in a clean marine environment in August 2015, coupling ground-based, in situ and remote sensing observations with Remotely Piloted Aircraft System (RPAS) and satellite observations (Sanchez et al., 2017). In this study, cloud droplet number concentration was not measured directly, and the closure study was conducted on cloud optical properties. RPAS measurements of cloud optical properties were more accurately reproduced by an ACPM simulation using a parameterization for entrainment compared to an adiabatic simulation. The present work is based on a analysis that is similar to Sanchez et al. (2017), and further extends aerosol-cloud closures with a sensitivity study on the impacts of aerosol and updraft on cloud optical properties.

Entrainment is well-known for influencing the boundary layer and clouds (e.g., Blyth (1993); Baker (1992); Carman et al. (2012)). Recent works have been published, investigating the role of entrainment and turbulence for broadening the cloud droplet spectra with an adiabatic parcel model (Grabowski and Abade (2017); Abade et al. (2018)), aiming to improve subgrid-scale representation for a large eddy simulation cloud model. Studies of entrainment-mixing mechanisms in cumulus clouds used manned aircraft observations (CIRPAS Twin Otter) to highlight the scale dependence of the mixing processes (Lu et al., 2018). However, as the scale of entrainment processes ranges from km to mm, Lu et al. (2014) point out the limitation of a 10 Hz sampling rate with a manned aircraft flying at 50 m s$^{-1}$ (spatial resolution 5 m). Similar conclusions were also deduced in Burnet and Brenguier (2007) for a resolution scale of 10 m (10 Hz data, manned aircraft Météo-France Merlin IV, NCAR C130) for turbulence and droplet evaporation. In Conant et al. (2004) and Meskhidze et al. (2005) closure studies on consistency between the variables $N_d$, the impact of entrainment was observed; however, the data was screened and only on the case studies close to adiabatic values were studied. Remotely Piloted Aircraft (RPA) bring new possibilities for studying aerosol-cloud interactions and optical cloud properties, as higher spatial resolution (e.i., 1.6 m with 10 Hz sampling rate) due to lower airspeed (16 m s$^{-1}$), which result in a better representation of the cloud.

This study focuses on an aerosol-cloud closure between in-cloud observations of downwelling solar irradiance from RPA and results of an ACPM initialized with RPA and ground-based measurements. The second section introduces the case study observed during the BACCHUS field campaign in Cyprus with a description of the ground-based observations of aerosol number size distribution and CCN, and airborne observations of temperature, relative humidity, vertical wind velocity, particle number and solar irradiance. The third section of this study focuses on the ACPM, and how a parameterization of entrainment-mixing is applied to the adiabatic simulation to take into account for the impact of entrainment. The last section highlights the closure on cloud optical properties with a sensitivity study that compares adiabatic profiles from ACPM simulations and the entrainment parameterization.





## 2 Cyprus case study

Cyprus is a highly relevant environment to study aerosols, particularly dust and ice-nucling particles (e.g., Schrod et al. (2017)), as the island is located in the Mediterranean Sea, at the intersection of pollution from Europe, Middle East, and dust from the Sahara. Cyprus is also impacted by marine aerosols and local anthropogenic emissions. The present study focuses on the

5 BACCHUS field campaign in Cyprus, which took place from 2015/03/05 to 2015/04/02. Ground-based instruments, remote-sensing, and RPAS activities contributed to the field campaign. 52 scientific flights were conducted with the RPAS platforms corresponding to 38 hours of airborne observations. This case study concentrates on one day flight measurements, and contains all of the necessary elements to study aerosol-cloud interactions by combining the RPA measurements with aerosol and CCN measurements at the ground. The purpose of this case study is to use in situ ground-based and airborne observations to initialize

an aerosol-cloud parcel model and compare in situ observations of cloud optical properties to those simulated in the ACPM. The present case study focuses on a RPA flight on 2015/04/01, which measured convective updrafts at cloud base and cloud optical properties of a stratocumulus layer (Fig.1).

### 2.1 Ground-based observations

Cyprus Atmospheric Observatory at Agia Marina Xyliatou (40 km West of Nicosia, 35.0386N; 33.0577E; 535 m a.s.l. (meters above sea level)) is operated by the Cyprus Institute, and provided complementary measurements of physicochemical properties of aerosols during the BACCHUS field campaign. Atmospheric studies including data from the Agia Marina Station in Cyprus have already been published on ozone concentration observations (Kleanthous et al., 2014), and particle matter variability (Pikridas et al., 2018). The station is part of the WMO-GAW regional station, EMEP et AERONET networks. Among the

instrumentation installed at the ground-based site, multiple measurement devices provided input to conduct an aerosol-cloud closure study. A miniature CCN instrument provides the number of activated particles at 0.24 % supersaturation (Roberts and Nenes, 2005). A scanning mobility particle sizer (Grimm 5400 SMPS) measures the aerosol number dry size distribution from 10 to 360 nm diameter. An optical particle counter (Grimm OPC 1.108) gives the number of particles per bin for dry sizes between 0.3 and 20 $\mu$m (14 bins). A condensation particles counter (CPC, model TSI 3010) counts the total aerosol

concentration (particle diameter $D_p$ > 10 nm) and is also used to normalize the SMPS measurements. An aerosol chemical speciation monitor (Q-ACSM, Aerodyn Research Inc) provides the chemical composition of non-refractory submicron aerosol particles. The ground-based measurements were conducted at a site that was 2 km from the RPAS operations.

### 2.2 Particle size distribution

Figure 2 shows the time series of the aerosol particle number distributions for 2015/04/01 measured at the Cyprus Atmospheric

Observatory, where the black rectangle represents the 3-hour period selected to average the aerosol size distribution and the magenta lines represents the time period of the RPA flight (take-off at 2:00 pm local time, 11:00 am UTC time). The aerosol particle number size distribution shows the presence of modes at 50 and 150 nm, with a trough near 100 nm implying cloud-




processed aerosol. Figure 3a presents the average size distribution from normalized SMPS, as well as ground-based and RPA OPC measurements. A minimum at 100 nm (known as the Hoppel minimum) is visible in Fig.3a. OPC concentrations of ground-based and RPA measurements (from surface to cloud base) are within a factor of two, which is within variability observed at the ground station on 2015/04/01. As the CPC measurements ended on 2015/03/27 (end of the BACCHUS field

campaign), no simultaneous measurements of aerosol number concentrations between the CPC and the SMPS are available for the case study day. Therefore, to quantify uncertainties between the integrated SMPS and CPC aerosol concentrations, CPC and SMPS data were compared for one week period (2015/03/20 to 2015/03/27). The CPC is a reference counting instrument and used to normalize the integrated SMPS concentration (Wiedensohler et al., 2012). To account for uncertainties associated with the SMPS inversion routines, we compare periods with and without new particle formation events, and the integrated

SMPS/CPC ratio shows a mean value of $0.63 \pm 0.16$ and $0.65 \pm 0.15$, respectively. Consequently, we use the minimum ratio (ca. 0.5) as the lowest concentration for the ACPM simulations (Section 3.2). On the time scale of hours, the inactivated CCN, or interstitial aerosol, do not change size or supersaturation $SS$ (Hudson et al., 2015). The cumulative distribution of particle number (based on SMPS and OPC measurements,Fig.3b) is used to estimate the number of particles that can grow into cloud droplets at a given diameter. In Fig.3, the Hoppel minimum diameter at 100 nm corresponds to an estimate of 400 cm$^{-3}$ particles

that activate to form cloud droplets. Similarly, based on the CCN measurements at the ground-station, the CCN concentration at 0.24 % $SS$ corresponds to 420 cm$^{-3}$. These results imply that a characteristic in-cloud supersaturation is close to 0.24 % $SS$. The critical diameter at 0.24 % $SS$ is 94.5 nm. Table 1 summarizes these parameters, diameters and concentrations.

## 2.3  RPAS observations

The RPAs are commercially available Skywalker X6 models that have been modified to be equiped with atmospheric mea-
20 surement instruments (Fig.4). The wingspan is 1.5 m long, and take-off weight varies between 1.5 kg and 2.5 kg depending on the mission specific payload. The RPA's autonomous navigation system is the open source autopilot Paparazzi from Ecole Nationale de l'Aviation Civile (Brisset et al., 2006). All the RPAs measured temperature (IST, Model P1K0.161.6W.Y.010), absolute pressure (All Sensors, Model 15PSI-A-HGRADE-SMINI), relative humidity (IST, P14 Rapid-W). Measurement er-
rors for the relative humidity and temperature sensors are $\pm$ 5 % and $\pm$ 0.5 °C, respectively. The RPAs had a video camera

attached to the wing (Camsports EVO PRO 2). The Centre National de Recherches Météorologiques (CNRM) group deployed different types of instrumented RPAs. Among them was an aerosol-RPA, equipped with an optical particle counter (OPC, Met One Model 212-2, for aerosol size between 0.3 and 3 $\mu$ m), and a wind-RPA, equipped with a 5-hole probe (Aeroprobe Corporation) and an Inertial Navigation System (INS, Lord Sensing Microstrain 3DM-GX4-45) to measure vertical wind ve-
locities near cloud base and pyranometers (LICOR LI-200R pyranometers, from 400 to 1100 nm wavelengths) to measure

cloud optical properties. The field to operate the RPAS (35.056429N; 33.055761E; 450 m a.s.l.) was located 1 km north of the Cyprus Atmospheric Observatory instrumented site. A rectangular airspace approximately 2.6 x 1.7 km$^2$ was used for the flight operations with a ceiling at 2286 m a.s.l.; 7500 ft a.s.l.



## 2.4 Case study : stratocumulus layer

The flight with the wind-RPA took place on 2015/04/01 at 11:00 am UTC (Flight 67) for duration of 1 hour and 20 min. The flight plan, as shown in Fig.1, consisted of a first set of 1.5 km straight-and-level legs at 1000 m a.s.l. near cloud base, then a profile up to 2100 m a.s.l. through the stratocumulus layer, and another set of straight-and-level legs at 950 m a.s.l.. Figure 5

shows the vertical wind distributions measured by the wind-RPA for the two sets of legs before and after the profile through the cloud layer. Even as the altitude of the legs were slightly different (1000 m a.s.l. and 950 m a.s.l.) because of an evolving boundary layer, nearly the same vertical wind velocity distributions are obtained before and after the cloud layer sampling. The similarity between the vertical wind velocity distributions demonstrate that the boundary layer dynamics were relatively constant throughout the flight and that the 5-hole probe functioned well even after a profile through the cloud layer. Comparing

Fig.5 with vertical wind distributions obtained during the BACCHUS field campaign at Mace Head Research Station in Ireland (Calmer et al. (2018); Sanchez et al. (2017)), it is noticeable that the vertical wind velocity distribution for the Cyprus case study is wider than the distributions obtained in Ireland (Cyprus: -2.5 < vertical wind < 4 m s$^{-1}$; Ireland: -1.5 < vertical wind < 2 m s$^{-1}$). The pictures in Fig.6 captured during the flight by the video camera show the cloud base and cloud top of the stratocumulus layer. By combining information between the video camera and the altitude of the wind-RPA, the history of

the flight is described in Table 2. Each period is also confirmed by pyranometer measurements (Fig.7). Broadband shortwave pyranometers mounted on the top and at the bottom of the RPA fuselage provided upwelling and downwelling profiles of solar irradiance. Normalized pyranometer profiles are shown without correction of the oscillations due to the cosine-angle response of direct sunlight on the sensor (Fig.7). These oscillations are particularly visible on the downwelling pyranometer above the cloud layer. Results highlight the fraction of shortware radiation of the incoming solar irradiance through the cloud layer. The

profiles of cloud measured optical properties from the RPA are compared in the next section with those of the ACPM. Figure 8 shows the Hysplit model (Stein et al., 2015) run for three days ending at 1000 m a.s.l. (altitude of cloud base) over the field site on 12:00 pm UTC 2015/04/01. The back trajectories show air masses originated from the Westen Mediterranean Basin, with trajectories carrying anthropogenic sources from Southern Europe, Northern Africa, and Turkey. The aerosol number concentrations are similar to regional urban background (Reddington et al., 2011) mixed with particles from recent particle

formation events and sea salt emissions.

## 2.5 RPA vertical profiles

Figure 9 presents ascent profiles of the atmosphere sampled by the wind-RPA during the flight. Profiles of temperature and relative humidity during the ascent and the descent of the RPA are similar, particularly in cloud. The temperature in the boundary layer decreases -10.1 °C km$^{-1}$, which is close to a dry adiabatic lapse rate. In cloud, the lapse rate changes to -4.5 °C

30  km$^{-1}$ (Fig.9a). The relative humidity increases from 75 % at the ground to 100 % at the cloud base (1020 m a.s.l.), and then decreases again closer to the cloud top (Fig.9b). As mentionned in Sanchez et al. (2017), measurements error for the relative humidity is ± 5 %, however, as the sensors are not accurate at RH > 90 %, the measured values have been scaled such that RH measurements are 100 % in a cloud (the air mass is considered saturated). The in situ measurements have been approximated





by linear expressions that serve as input parameters for the ACPM in Section 3.2 (magenta lines in Fig.9). The profile of equivalent potential temperature, which is conserved for changes in the air parcel pressure in Fig.9c, shows a neutrally buoyant layer below the cloud base, which implies a well-mixed boundary layer. In addition, profiles of aerosol number concentrations (Fig.10) are measured during an earlier flight on the same day (Flight 65, aerosol-RPA, 8:50 am UTC, 11:50 am local time),

and present similar concentrations in the atmospheric boundary layer from the ground to cloud base (1020 m a.s.l.). This observation also confirms a well-mixed boundary layer, such that ground-based CCN and aerosol size distributions are then representative of the aerosol concentrations at cloud base.

## 3 Aerosol-Cloud Parcel Model

The term closure is used in a number of aerosol-cloud interactions studies to evaluate the cloud droplet number concentration

$N_d$ obtained from a parcel model based on observations of aerosol and updraft velocities (Conant et al. (2004); Fountoukis et al. (2007); Kulmala et al. (2011)). In the present work, as no direct measurements of $N_d$ were available, the closure is addressed through the in-cloud fraction of transmitted shortwave radiation profile deduced from the ACPM and measured with the pyranometers. The ACPM is used as a proxy for cloud droplet number concentration. An entrainment parameterization is implemented on the model results to obtain a better agreement between the model and observations.

### 3.1 Description of the Aerosol-Cloud Parcel Model (ACPM)

The Aerosol-Cloud Parcel Model (ACPM) is based on Russell and Seinfeld (1998) and Russell et al. (1999), where the main equations explicitly described the processes of activation of aerosol particles and the condensation of water vapor on the resulting cloud droplets. The ACPM uses a fixed sectional approach for distinct aerosol populations to calculate particle growth under supersaturated conditions (Russell and Seinfeld, 1998). Evaporation from the entrainment is parameterized and applied

to the ACPM results. Deposition is also included but negligible for the study here. Droplet collision, coalescence and drizzle rates are negligible for the simulated values of liquid water content and cloud droplet number concentration in the present case (i.e. droplet diameter $D_p < 20$ nm). The model is designed to be initialized from aircraft-based field observations, and employs a dual-moment (number and mass) algorithm to calculate the particle growth. The aerosol population observed in our studies is assumed to be internally mixed as the particles generally undergo long-range transport from their source (Fig.8). However,

liquid water is treated in a moving section representation to have agreement between the particle number and mass (Russell and Seinfeld, 1998). To simulate drops in the model, aerosol particles are divided into 70 bins that are log spaced. The minimum size is 0.02 $\mu$m, and the maximum size is 3.0 $\mu$m. The equation describing the evolution of the thermodynamic energy of the air parcel is given by the vertical temperature gradient :

$$dT = -\frac{gwdt + Ldq_l}{c_p} \qquad (1)$$

where $dT$ is the change in temperature corresponding to the $dt$ time step in the ACPM, $w$ is the updraft velocity, $g$ is the acceleration due to gravity, $L$ is the latent heat of water condensation, $q_l$ is the liquid water mixing ratio, and $c_p$ is the





specific heat of water. Equation 1 relates the vertical wind velocity with the release of latent heat of a rising air parcel in an adiabatic parcel of air. The vertical velocities (updraft and downdraft) are measured near the cloud base or within the cloud. For the 1D model, positive updrafts generate supersaturated conditions in which aerosol particles are activated into cloud droplets. Therefore, the downdrafts are not considered in the simulation. The measured temperature profile is used to

parametrize entrainment. To apply the cloud-top mixing, a fraction of air at cloud base and a fraction of air above cloud top are mixed, conserving the total water content and the equivalent potential temperature (Sanchez et al., 2017). The gradient in the conserved variable is nearly linear, and is then used to adjust the liquid water content by assuming inhomogeneous mixing. The fraction of air masses originating from below and above the cloud layer is determined as :

$$\theta_{e,c}(z) = \theta_{e,ent} X(z) + \theta_{e,CB}(1 - X(z)) \tag{2}$$

where $\theta_{e,c}(z)$ is the equivalent potential temperature in cloud as a function of height, $\theta_{e,ent}$ is the equivalent potential temperature of the cloud-top entrained air, $\theta_{e,CB}$ is the equivalent potential temperature of air at cloud base, and $X(z)$ is the fraction of cloud-top entrained air as a function of height (Sanchez et al., 2017). Then, the entrainment fraction $X(z)$ is given by :

$$X(z) = \frac{\theta_{e,c}(z) - \theta_{e,CB}}{\theta_{e,ent} - \theta_{e,CB}} \tag{3}$$

Sanchez et al. (2017) illustrates the importance of including entrainment to simulate cloud optical properties using the ACPM. A similar approach for this case study is presented in the following sections.

### 3.2   Model inputs from ground measurements and RPAs

To initiate the APCM model, in situ measurements of aerosol size distribution and calculated hygroscopic properties from the ground-station are combined with vertical profiles of temperature and relative humidity and vertical wind velocity distributions

from the RPA. The aerosol size distribution described in Fig.3, along with the hygroscopicity parameter obtained in Section 3.3 are implemented in the ACPM to approximate the CCN spectra at cloud base. The temperature and humidity profiles (Fig.9, magenta lines) derived from observations of the RPA profile are used as input parameters in the ACPM model. Yet, in the cloud, the temperature and supersaturation are calculated. The ACPM temperature profile in the cloud is moist adiabatic.

In the literature, either a characteristic updraft velocity or an updraft distribution is used in ACPM. Conant et al. (2004), Hsieh et al. (2009), and Sanchez et al. (2017) have shown that a distribution of updraft velocities and a weighted distribution of in-cloud supersaturations better reproduce cloud microphysical properties than a single-updraft approximation. Consequently, positive vertical wind velocity distribution near cloud base are used as model input (updraft velocities from 0.1 to 4 m s$^{-1}$ in Fig.5). The APCM model simulates the cloud droplet growth using 40 bins of vertical wind velocities between 0 and 4 m s$^{-1}$

(Sanchez et al., 2016). Each bin corresponds to a maximum supersaturation and a number of CCN activated into cloud droplets.





The overall cloud microphysical properties are weighted based on the vertical wind velocity distribution. The cloud droplet number concentration corresponds to the summation of the number of CCN activated weighted with updraft, and is expressed as:

$$N_d = \sum_{i=1}^{40} f(w_i).N_{CCN}(w_i).w_i \tag{4}$$

Where $N_d$ is the cloud droplet number, $i$ is the bin number, $f(w_i)$ is the occurence of updraft $w_i$ at the supersaturation $Sc_i$, and $N_{CCN}(w_i)$ is the number of activated particles based on the cloud droplet distribution for $Sc_i$ and $w_i$. $N_d$ is volume weighted by the factor $w_i$. Results are in line with the case studies (marine environment) presented in Sanchez et al. (2017).

### 3.3   Aerosol-CCN closure

To describe the relationship between particle composition and CCN activity, Petters and Kreidenweis (2007) define the hygroscopicity parameter, $\kappa$, based on the Köhler theory (Seinfeld and Pandis, 2006). The hygroscopicity parameter, $\kappa$, represents a quantitative measure of water-soluble ions on CCN activity.

$$\kappa = \frac{4A^2}{27D_p^3 \ln^2 S_c} \tag{5}$$

where $D_p$ is the droplet diameter, $S_c$ is the critical supersaturation, and $A$ is expressed as:

$$A = \frac{4\sigma_w M_w}{RT\rho_w} \tag{6}$$

where $M_w$ is the molecular weight of water, $\sigma_w$ is the solution surface tension, $\rho_w$ is the water density, $R$ is the universal gas constant, and $T$ is the temperature. $\kappa$ calculated using the CCN measurement with a critical dry diameter at 100 nm for a supersaturation of 0.24 % gives 0.3. The value of $\kappa$ calculated from the aerosol size distribution and CCN measurement is compared to $\kappa$ obtained from chemical constituents measured by the ACSM instrument at the ground station (Fig.11). From the

ratio provided by the ACSM approximated to 50 % ammonium sulfate/organic matter submicron aerosol, $\kappa$ is estimated to be 0.26. The sulfate are assumed to be in the form of ammonium sulfate, and the organic matter (or insuluble fraction) presented a hysgroscopicity of 0.1 based on typical values of observed organic hygroscopicity (Petters and Kreidenweis (2007), Gunthe et al. (2009), Prenni et al. (2007)). These values are in good agreement, and confirm a closure between aerosol physical and chemical properties and the CCN measurements.

### 4   Aerosol-cloud closure study

The purpose of the parcel model is to serve as the link between in situ measurements of aerosol and vertical velocity distributions to the observed cloud microphysical properties. 1D aerosol-parcel models with explicit cloud microphysics are



specifically designed to explore droplet growth/evaporation for a given CCN spectra and vertical velocity distribution. The procedure is to run the 1D model adiabatically, then use the observations of mixing of the conservative variable to calculate how much water should have evaporated due to cloud entrainment. (Fig.12a). Figure 12a presents the water vapor content derived from the relative humidity ($q_v$), which is equivalent to the total water content ($q_t$) above and below the cloud, as a

5 function of equivalent potential temperature. The total water content and equivalent potential temperature in an adiabatic parcel is conserved, however in Fig.12a the total water content decreases from 7.5 g kg$^{-1}$ at cloud base to 6.7 g kg$^{-1}$ cloud top ($R^2 = 0.95$). An adiabatic profile would show that the total water content at cloud top would remain unchanged from the cloud base value of 7.5 g kg$^{-1}$. This indicates that the cloud is not adiabatic. The total water content at cloud top is much lower than the total water content closer to cloud base, suggesting air masses above the cloud top are the source of dry air entrainment,

consistent with previous studies of stratocumulus cloud top entrainment (Wood, 2012). The decrease in water vapor content throughout the cloud is a result of the combination of cloud top entrainment of dry, warm air and water vapor condensation. In-cloud measurements of equivalent potential temperature are reliable despite the presence of liquid water. Using Eq.2 and measurements of the equivalent potential temperature throughout the cloud, the fraction of entrained air can be estimated and the in-cloud profile of liquid water content can be calculated. The linear relationship between the simulated total water content

and measured equivalent potential temperature is a result of the cloud reaching a steady state, with air coming from cloud base and cloud top (Fig.12a). The reduction in number concentration due to entrainment is driven by the amount of evaporated water as we approximate the evaporation through inhomogeneous mixing (Jacobson et al., 1994). Figure 12b presents the profiles of LWC calculated from the ACPM in case of the adiabatic simulation and when the entrainment parameterization is considered.

## 4.1 Cloud droplet number concentration

Results of the ACPM for the profile of cloud droplet number concentration ($N_d$) and effective radius are presented in Fig.13. For the adiabatic reference case, $N_d$ is around 420 cm$^{-3}$. The adiabatic profile of $N_d$ is compared to the profile incorporating the entrainment parameterization that forces the model to the observed temperature lapse rate (Eq.2). Most of the closure studies neglect entrainment (Snider et al. (2003); Conant et al. (2004); Peng et al. (2005)), as they investigated aerosol closure,

and observed that the entrainment did not affect much the results at cloud base. However, for the case studies at Mace Head (Sanchez et al., 2017), the difference between observed and simulated parameters (in the case, the cloud-top temperature) suggested a source of heating in the cloud, and a closer approximation of cloud radiative properties was obtained when the entrainment was included in the model results. The entrainment parameterization approximates the impact of inhomogeneous mixing on $N_d$ due to evaporation of a subset of the cloud droplet population. For the entrainment case, $N_d$ reaches highest

number concentration a few tens of meters above cloud base, and then decreases with altitude, as the inhomogeneous mixing is assumed (Fig.13a). However, $N_d$ is very sensitive to the entrainment fraction at cloud base, as the droplets are very small so even a small change in the amount of water evaporated (from entrainment) will cause a large difference in the number concentration. Shaded areas in Fig.13a highlight the model sensitivity to a small variation of LWC in obtaining cloud droplet number, as the sensitivity of the $N_d$ profile is a function of LWC. This variation in LWC is obtained based on the mixing





line (Fig.12a) and represents the standard deviation calculated from the difference between the mixing line in cloud and its best fit (0.052 g kg$^{-1}$). $N_d$ in the adiabatic profiles varies within ± 160 cm$^{-3}$ near cloud base (ca. 45 % variation relative to the adiabatic reference case). However, variations up to 230 cm$^{-3}$ are observed for the entrainment profiles near cloud base (ca. 230 % variation relative to the entrainment reference case). Yet, higher in cloud, the impact of LWC variation on $N_d$ is

5 less pronounced. The peaks of $N_d$ for the entrainment profile are then sensitive to observed temperature profiles; however, as clouds are optically thin at cloud base, the impact of this sensitivity on overall cloud optical properties is small. Yet, at cloud top, the maximum difference in $N_d$ between the entrainment and adiabatic ACPM profiles is $\sim$ 300 cm$^{-3}$, which ultimately, plays a large role in the overall cloud optical properties. Profiles of direct observations of cloud droplet numbers show a similar large sensitivity at cloud base and a decrease in number with altitude (Roberts et al. (2008); Rauber et al. (2007)).

## 4.2 Cloud optical properties

To study the cloud optical properties, solar irradiance obtained from the pyranometers mounted on the wind-RPA is compared to ACPM fraction of transmitted shortwave radiation profiles, which represents the solar irradiance transmission through the cloud layer. The transmission through the cloud layer is approximated by downward integration of the calculation of albedo and subtracting from unity. For example, an infinitely thin cloud has an albedo of zero; therefore, 100 % of incoming solar

irradiance is transmitted through the cloud. As the cloud thickens, the albedo approaches unity meaning that all incoming solar irradiance is reflected back to space (Fig.14). To derive the cloud optical properties from the ACPM, the method presented in Sanchez et al. (2017), is followed here, based on Hansen and Travis (1974) and Stephens (1978). The cloud droplet extinction is proportional to the total droplet surface area and has the form :

$$\sigma_{ext} = \int\limits_0^\infty Q_{ext}(r)\pi r^2 n(r)dr \tag{7}$$

where $r$ is the radius of the droplet, $n(r)$ is the number of the cloud droplets with a radius of $r$, and $Q_{ext}(r)$ is the Mie efficiency factor. $Q_{ext}(r)$ asymptotically approches 2 for water droplets at large size ($r > 2$ $\mu$m; Seinfeld and Pandis (2006)). The cloud optical depth is defined as:

$$\tau = \int\limits_0^H \sigma_{ext}(h)dh \tag{8}$$

where $H$ is the cloud thickness and $\sigma_{ext}$ is the cloud droplet extinction calculated from the simulated cloud droplet size

distribution (Eq.7). The cloud albedo is then calculated with $\tau$:

$$albedo = \frac{\sqrt{3}(1-g)\tau}{2+\sqrt{3}(1-g)\tau} \tag{9}$$





with $g$ the asymmetric scattering parameter. The albedo is estimated based on the cloud optical depth and the asymmetric scattering parameter (approximated as 0.85 based on the Mie scattering calculation).

The solar irradiance profile from the RPA, based on the normalized downwelling pyranometer measurements during the descent, is used to compare simulated and observed cloud optical properties (Fig.14). To facilitate comparison with the model results, normalized pyranometer is averaged every 50 m (which averages the oscillations related to pitch-and-roll cosine-angle response of the pyranometer). Observations show a sharp gradient in the attenuation of downwelling solar irradiance near cloud top and decrease to ca. 0.2 at cloud bottom. Overlaid on Fig.14 are model results from the ACPM for adiabatic and entrained cases. In order to compare ACPM and RPA observations, albedo of the cloud layer is calculated top-down using the

profiles of simulated cloud droplet number and size distribution in Fig.13 to estimate the amount of solar irradiance reflected back to space and subtracted from unity to compare with the downwelling pyranometer profile. The mean difference in the fraction of transmitted shortwave radiation for in situ measurements and adiabatic simulation is 0.3, although when accounting for entrainment, the mean difference is only 0.03. Therefore, comparison between RPA observations and ACPM for adiabatic and entrainment fraction of transmitted shortwave radiation profiles suggests that cloud optical properties are best represented

when including entrainment mixing of cloud-top air.

## 4.3    Sensitivity study on cloud optical properties

In addition to comparing ACPM results between entrainment and adiabatic cases, a sensitivity analysis presented here explores the impact of a change in aerosol particle number concentrations as well as changes in the vertical velocity distribution

on the cloud optical properties. Profiles of the cloud droplet number and effective radii (Fig.13) and cloud optical properties (Fig.14) are also simulated with the inputs of aerosol number concentration multiplied by two ($2N$) and the updraft velocity distribution divided by two ($w/2$). Increasing the aerosol concentrations by a factor of two results in an aerosol concentration of $\sim$ 2400 cm$^{-3}$ representing more polluted conditions. Such an increase in aerosol/CCN concentrations also increases cloud droplet number concentration (Fig.13a), decreases the effective radii (Fig.13b), and presents a cloud with a higher albedo. In

addition, halving the vertical wind velocity distribution results in a distribution with maximum vertical wind velocities near 2 m s$^{-1}$, which also happen to be similar to the updraft velocities observed in marine stratocumulus cloud layers over Mace Head Research Station, Ireland (Calmer et al., 2018). The lower vertical wind velocities also results in lower cloud droplet number concentrations and larger effective radii owing to lower in-cloud supersaturations (Fig.13) . The lower cloud droplet number and smaller effective radii results in lower albedo of the cloud layer (Fig.14). A decrease of 70 cm$^{-3}$ in cloud droplet number

is observed when the updraft velocity distribution is divided by two ($dw$); and an increase of 50 cm$^{-3}$ droplet number occurs when the number of dry particles is multiplied by two ($dN$). Factor of two changes in updraft distribution causes the fraction of transmitted shortwave radiation to decrease by 0.003 in the adiabatic case, and 0.005 in the entrainment case, corresponding to an increase in extinction. Likewise, a factor of two increase in aerosol size distribution leads to a -0.002 (adiabatic case) and -0.004 (entrainment case) changes in the fraction of transmitted shortwave radiation through the cloud (corresponding to





similar net changes in cloud albedo, Fig.14). To summarize, variations of $N$ and $w$ correspond to a change in a range between -0.004 and 0.005 in the fraction of transmitted shortwave radiation. Yet, the change in the fraction of transmitted shortwave radiation between adiabatic and entrainment cases is 0.15, corresponding to a factor of three change in cloud albedo compared to changes in droplet number and vertical wind velocity distributions.

The sensitivity of albedo to changes in droplet concentrations was first introduced by Platnick and Twomey (1994), which defined a degree of susceptibility function of cloud optical thickness, effective radius, and liquid water content. Clouds formed in cleaner environments are likely to be of higher susceptibility compared to clouds in polluted areas, which illustrate the link between pollution and cloud albedo proposed by Twomey (1977). Painemal and Minnis (2012) used the same definition of

susceptibility to investigate the albedo sensitivity to changes in the cloud microphysics. The increase of albedo susceptibility with LWC was observed for three maritime clouds regimes. Feingold (2003) and McFiggans et al. (2006) used the equation $S(X_i) = dlnY/dlnX_i$ as a representation of the sensitivity of $X$ on $Y$. $Y$ is a physical property of the cloud (e.g. the effective radius, the cloud droplet number concentration) and $X$ is a meteorological parameter (e.g. updraft velocity, LWC) or property of the dry aerosol (e.g. concentration, size distribution). A similar calculation of sensitivity is used to compare the influence of

the particle number and updraft on albedo in the adiabatic and entrainment case.

$$S_{di} = \frac{dlna_{ref} - dlna_{di}}{dlnNd_{ref} - dlnNd_{di}} \qquad (10)$$

Where $a$ is the albedo at cloud top, $N_d$ is the cloud droplet number at cloud top, $ref$ represents the reference case, and $di$ represents either a variation of the particle number $dN$ or the updraft velocity $dw$ in the adiabatic or the entrainment case. Table 3 summarizes the input values for the sensitivity calculation and results are presented in Table 4. The sensitivity in

the reference case between the adiabatic and entrainment cases (0.118) is higher than the other sensitivities, demonstrating a significantly larger importance of the entrainment parameterization on albedo compared to the initial conditions of particle number $N$ or updraft $w$. Figure 15 shows the calculation of $S_{di}$ as a function of the cloud depth. As mentioned in Section 4.1, the initial conditions influence mainly the cloud base, and then, higher in cloud, albedo is more sensitive to the entrainment parameterization.

## 5   Conclusions

An aerosol-cloud closure on cloud optical properties is conducted on a case study by comparing measured and simulated shortwave radiation transmission profile. The measurements were conducted for this closure study on one day (2015/04/01) of the one-month BACCHUS field campaign in Cyprus. Ground-based measurements at Cyprus Atmospheric Observatory are combined with Remotely Piloted Aircraft (RPA) observations to initiate an Aerosol-Cloud Parcel Model (ACPM) to compare

observed and simulated cloud optical properties. Input parameters of the model include the ground-based aerosol size distribution obtained from combined SMPS and OPC distributions averaged for the studied period as well as the vertical velocity





distribution at cloud base as measured by the RPA. Vertical profiles of temperature and relative humidity measured during a RPA flight are implemented in the model. The in-cloud lapse rate is lower than simulated for adiabatic conditions, suggesting cloud-top mixing from above the stratocumulus layer. Two different simulation cases are studied with the ACPM (i.e., an adiabatic case and an entrainment case), where the in-cloud temperature profile is taken into account to calculate the fraction of

cloud-top entrained air throughout the cloud. The adiabatic ACPM simulations yield cloud droplet number concentrations (ca. 400 cm$^{-3}$) that are similar to those derived from the Hoppel minimum analysis (420 cm$^{-3}$). Cloud optical properties have been observed using the transmitted shortwave radiation profile measured by a downwelling pyranometer. The normalized transmitted shortwave radiation is then compared to simulations from the ACPM, and shows a better agreement with the entrainment parameterization rather than with the adiabatic profile. These results highlight the importance of accounting for entrainment in

deriving cloud optical properties.

To better evaluate the sensitivity of the ACPM results, variation of input parameters are implemented by multiplying the aerosol concentrations by two (from 1234 to $\sim$ 2400 cm$^{-3}$; even more polluted conditions), and dividing the updraft velocity distribution by two (maximum $w$ from 4 to 2 m s$^{-1}$; conditions similar to marine environment, (Lu et al. (2007), Calmer et al.

(2018)). A doubling of $N$ increases the maximum cloud droplet number by 50 cm$^{-3}$, whereas a reduction in $w$ decreases the maximum cloud droplet number by 70 cm$^{-3}$. The impact on cloud effective radius is relatively small, less than $\pm$ 1 $\mu$m changes in the radius (< 7 % in relative changes). These changes in cloud droplet number concentrations by varying $N$ and $w$ lead to changes between -0.004 and 0.005 in the fraction of transmitted shortwave radiation. In comparison, the change in fraction of transmitted shortwave radiation and albedo related to entrainment is 0.15. The sensitivity calculation $S_d$ of

albedo to cloud droplet number concentration shows the significant impact of entrainment-mixing compared to those of aerosol concentration and updraft velocity for cloud optical properties. These results are in agreement with the conclusion of closure studies conducted at Mace Head Research Station (Sanchez et al., 2017), whereby the incorporation of a parameterization for entrainment improves the estimate for shortwave radiative flux. The case studies in Cyprus (this study) and at Mace Head illustrate the significance of the entrainment processes in determining cloud optical properties in two different environments,

and highlight notably that entrainment processes reduce the impact of cloud radiative forcing. More observations in climatically different regions are needed to understand the relative impact of aerosol, updraft and entrainment on cloud radiative properties.

*Author contributions.*  The analysis of the case study was conducted by Radiance Calmer and Kevin Sanchez with the supervision of Greg Roberts. The coauthors have contributed to the writing of the manuscript. Greg Roberts and Radiance Calmer carried out the RPAS flights. The ACPM was run and interpreted by Kevin Sanchez. Karine Sellegri and David Picard operated the SMPS, and CPC instruments at the

Cyprus Atmospheric Observatory and provided the associated data. Greg Roberts operated the miniature CCN counter at the Cyprus Atmospheric Observatory. Michael Pikridas operated the ACSM and was responsible for the ground site at the Cyprus Atmospheric Observatory. Mihalis Vrekoussis was in charge of the RPAS regulations in Cyprus during the field campaign. Jean Sciare hosted the BACCHUS field campaign on behalf of the Cyprus Institute.



*Acknowledgements.* The research leading to these results received funding from the European Union's Seventh Framework Programme (FP7/2007-2013) project BACCHUS under grant agreement n°603445. The RPAS used for the experiments presented in this work have been developed by the Ecole Nationale de l'Aviation Civile (ENAC). The authors thank the ChArMEx project supported by ADEME, CEA, CNRS-INSU and Météo-France through the multidisciplinary program MISTRALS. The authors also thank Lynn Russell, from Scripps Insitution of Oceanography, University of California, San Diego, CA, for making the Aerosol-Cloud Parcel Model (ACPM) available for this study.





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



**Table 1.** Particle number distribution from SMPC and OPC during the 3-hour period including the flight. The values are obtained from Fig.3. The minimum and maximum values are based on the standard deviation of the cumulative sum of number of particles $N$

| total number of particles ($cm^{-3}$) | Hoppel minimum diameter (nm) | particle number at the Hoppel minimum ($cm^{-3}$) | number of particles at 0.24 % $SS$ ($cm^{-3}$) | diameter at 0.24 % $SS$ (nm) |
|---|---|---|---|---|
| 1234 ($\pm$ 63.6) | 100 | 400 (min=365.4 max=434.2) | 420 | 94.5 (min=85.99 max=103.95) |

**Table 2.** Profile history from cloud base (1000 m a.s.l.) to the ceiling (2100 m a.s.l.) and down near cloud base again (950 m a.s.l.) during the flight

| Video time (min) | Altitude (m a.s.l.) | Observations |
|---|---|---|
| 30:44 | 1072 | cloud base, start of the ascent profile |
| 32:55 | 1295 | change in visibility, enter in the cloud |
| 36:17 | 1602 | cloud top |
| 39:40 | 1904 | below a convection cell |
| 41:00 | 2102 | maximum altitude of the profile |
| 46:43 | 1730 | cloud cell, video-camera sees in cloud |
| 51:42 | 1121 | first sight of ground |
| 53:11 | 996 | cloud base, end of the descent profile |

**Table 3.** Input values to calculate the sensitivity $S_d$ used in Eq.10

| | adiabatic | | | entrainment | | |
|---|---|---|---|---|---|---|
| | ref | dN | dw | ref | dN | dw |
| Albedo* | 0.917 | 0.919 | 0.915 | 0.767 | 0.771 | 0.762 |
| Nd* ($cm^{-3}$) | 436.4 | 503.1 | 367.0 | 96.3 | 111.0 | 81 |
| LWC* (g $m^{-3}$) | 1.47 | | | 0.324 | | |

*at cloud top



**Table 4.** Results of the sensitivity $S_d$ of albedo for cloud droplet number concentration with variation of aerosol concentration (dN) and updraft velocity (dw) given by Eq.10. The comparison of sensitivity calculation between adiabatic and entrainment cases at cloud top is $S_{d_{ent}}$=0.118

|          | adiabatic | entrainment |
| -------- | --------- | ----------- |
| $S_{dN}$ | 0.012     | 0.032       |
| $S_{dw}$ | 0.018     | 0.037       |





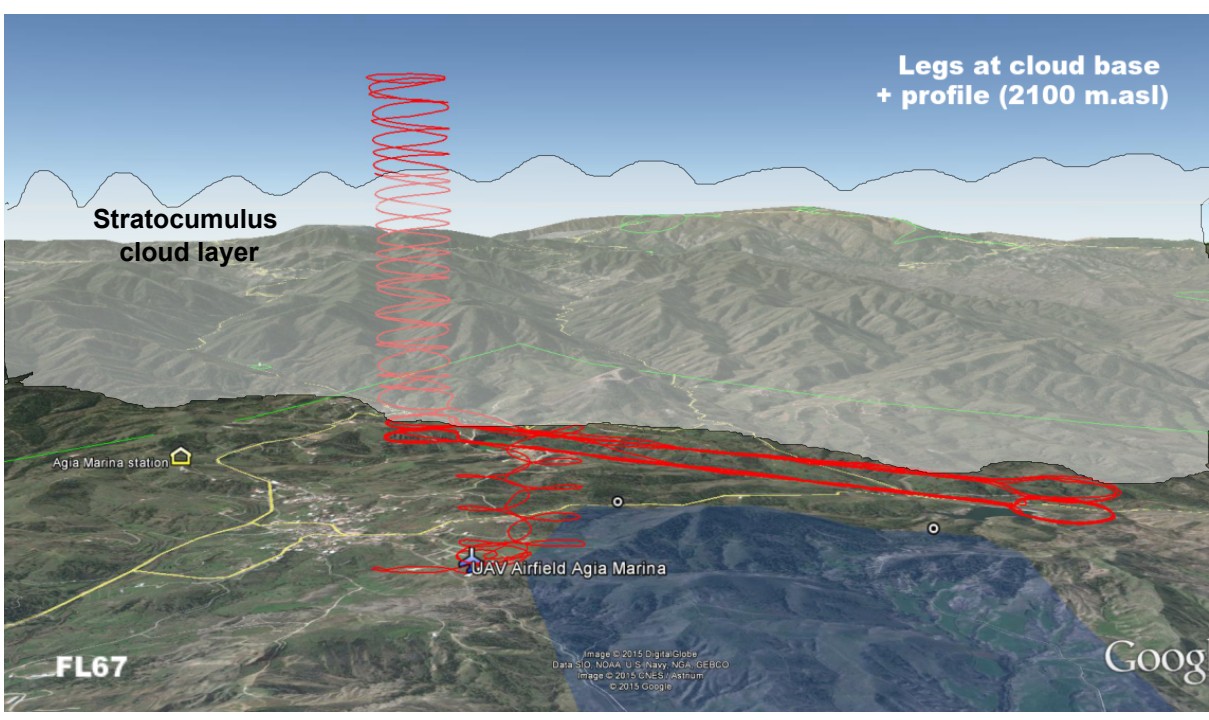

**Figure 1.** Flight plan for the flight (Flight 67), legs at 1000 m a.s.l., profile up to 2100 m a.s.l. and legs at 950 m a.s.l.. The approximate location of the stratocumulus layer is overlaid on the flight track.





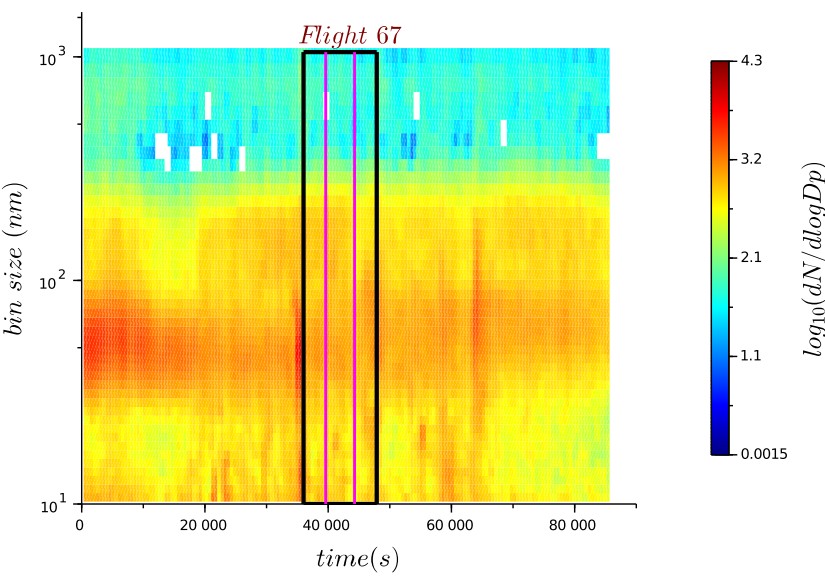

**Figure 2.** Contour plot showing time series of SMPS data 2015/04/01. The black rectangle represents the selected data for the analysis (3 hours) and the magenta lines delimit the flight (Flight 67).





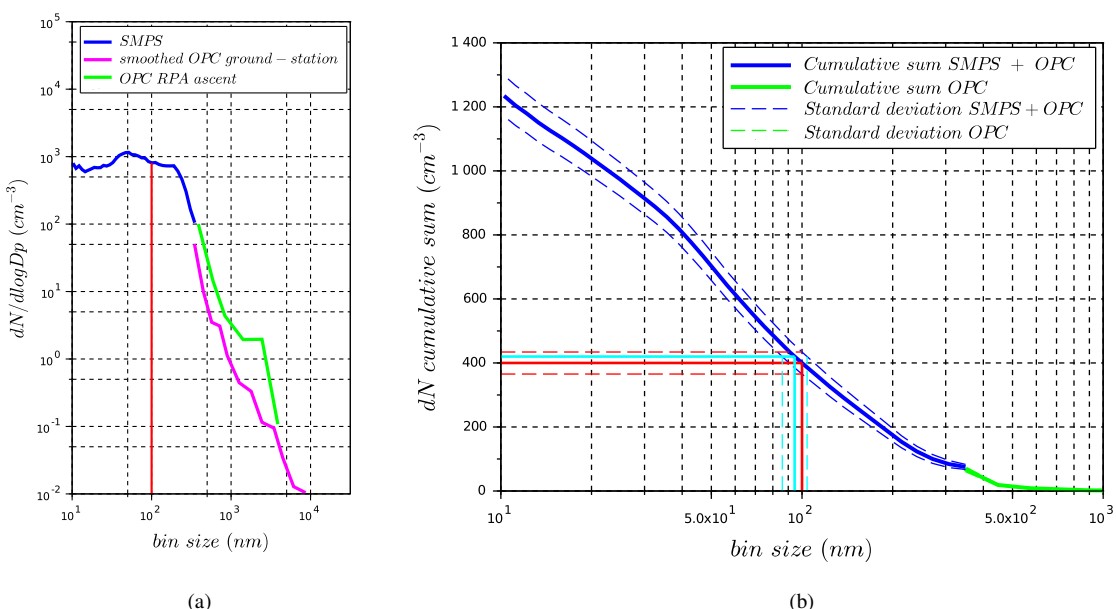

| (a) | (b) |
|-----|-----|

**Figure 3.** a) Particle size distribution showing combined data from the SMPS (blue), the ground-based OPC (magenta) and the RPA-OPC (green). The red line indicates the Hoppel minimum diameter. b) Cumulative particle size distribution with a combination of data from the SMPS and the ground-based OPC. The red solid lines indicate the number of particles at the Hoppel minimum diameter (100 nm), the red dotted lines correspond to the associated number of particles for the standard deviation of the cumulative sum. The cyan solid lines correspond to the aerosol diameter for 420 cm$^{-3}$ particles ($SS = 0.24$ %). The cyan dotted lines indicate the aerosol diameters for the standard deviation of the cumulative sum. The values are written in Table 1.


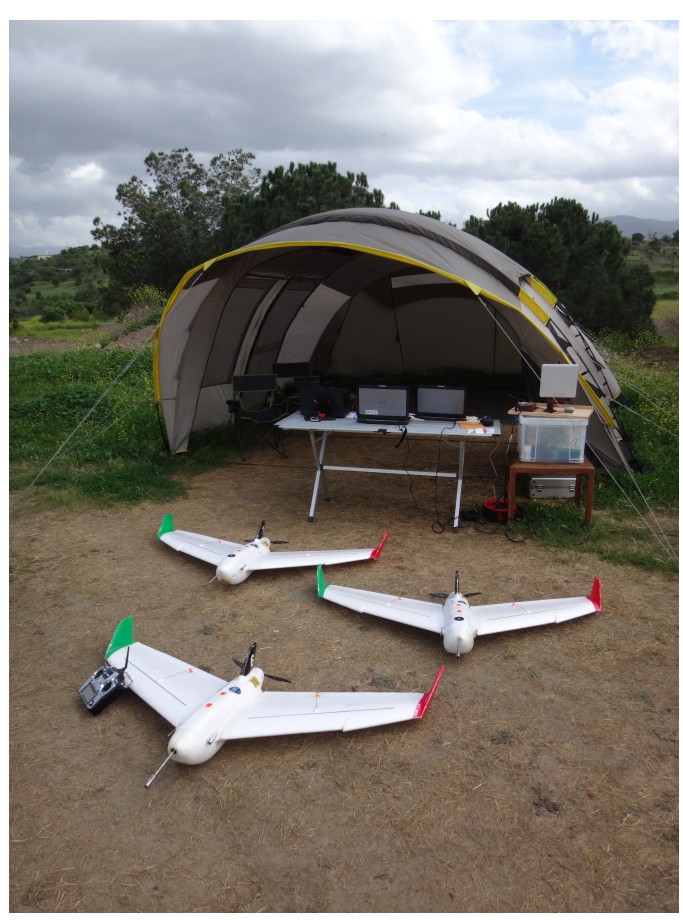

**Figure 4.** Remotely Piloted Aircraft on the operation field during the BACCHUS field campaign in Cyprus (March 2015).

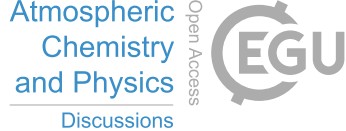

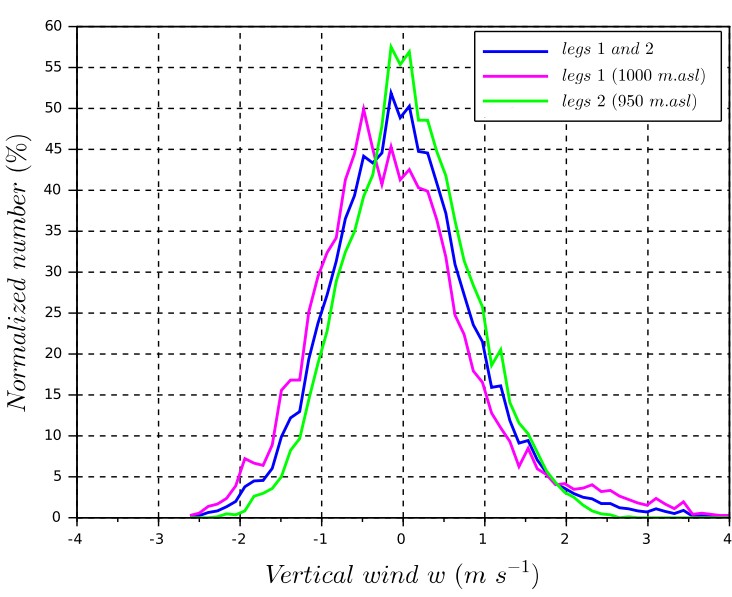

**Figure 5.** Vertical wind distributions from straight-and-level legs near cloud base during the flight.



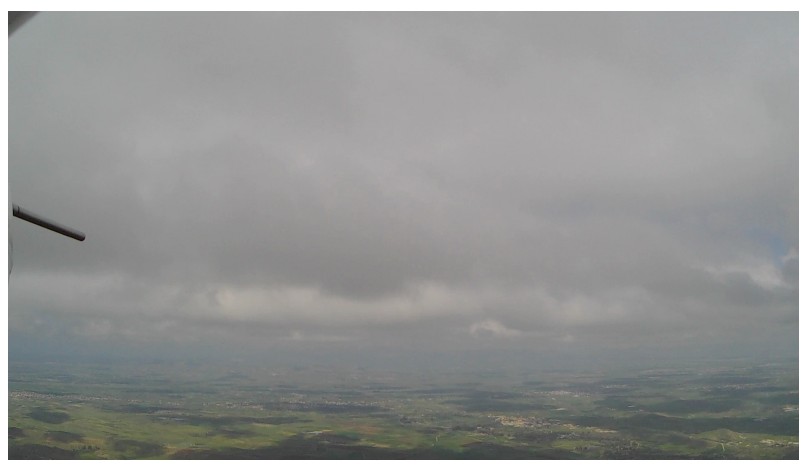

(a)

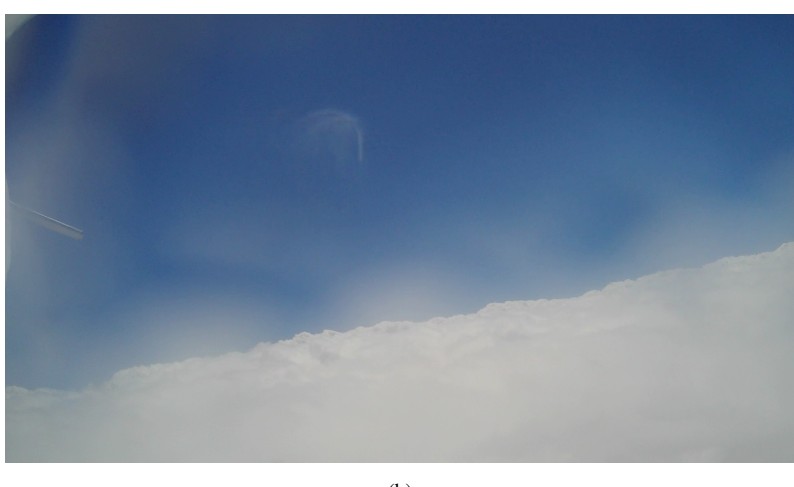

(b)

**Figure 6.** Pictures of the flight from the on board video camera (Table 2), a) near cloud base (1000 m a.s.l.), b) above clouds (2000 m a.s.l.)





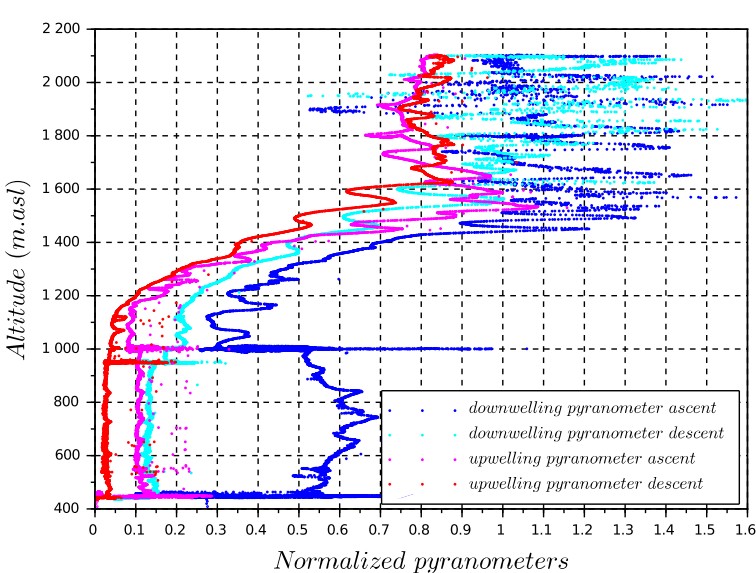

**Figure 7.** Solar irradiance profiles from normalized pyranometer measurements during the flight. The normalization parameter is the measured solar irradiance for clear sky, above the cloud layer, validated from the pyranometer profiles and the video camera (above 1600 m a.s.l.)





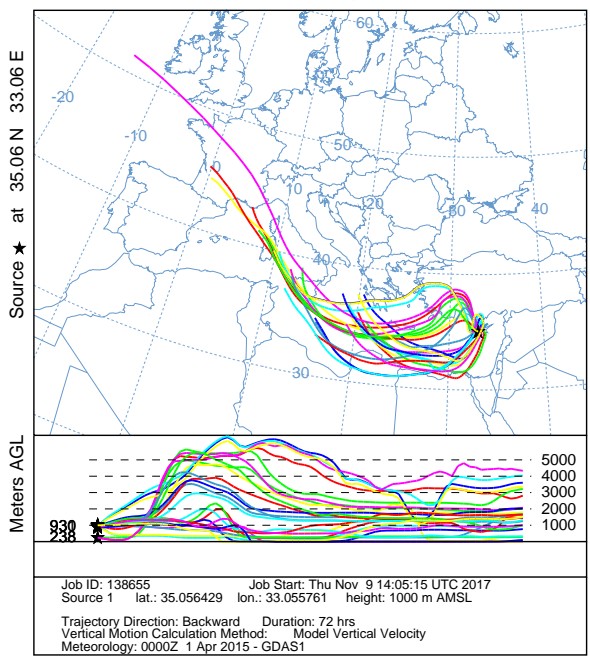

**Figure 8.** Hysplit model showing 3-day backtrajectories 1 April 2015. Black Star shows the location of the ground station and RPA operations near Agia Marina Xyliatou in Cyprus.





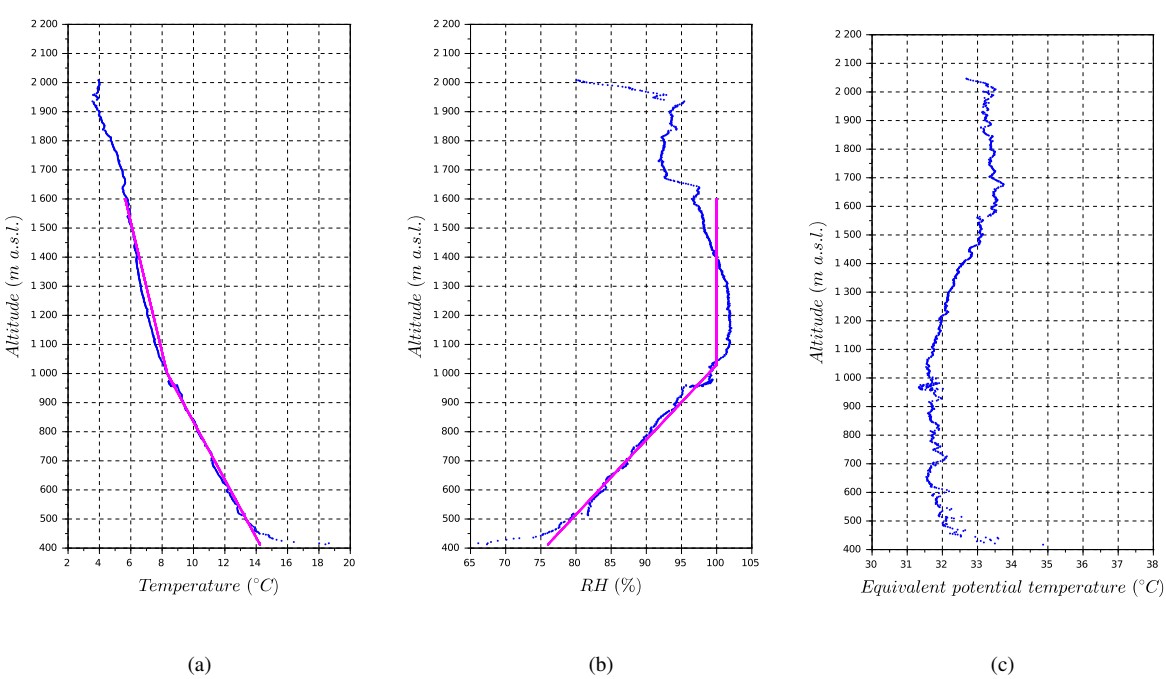

(a)  (b)  (c)

**Figure 9.** Meteorologic profiles during the flight a) temperature, b) relative humidity. The magenta curves correspond the linear best fit implemented in ACPM. c) Equivalent potential temperature





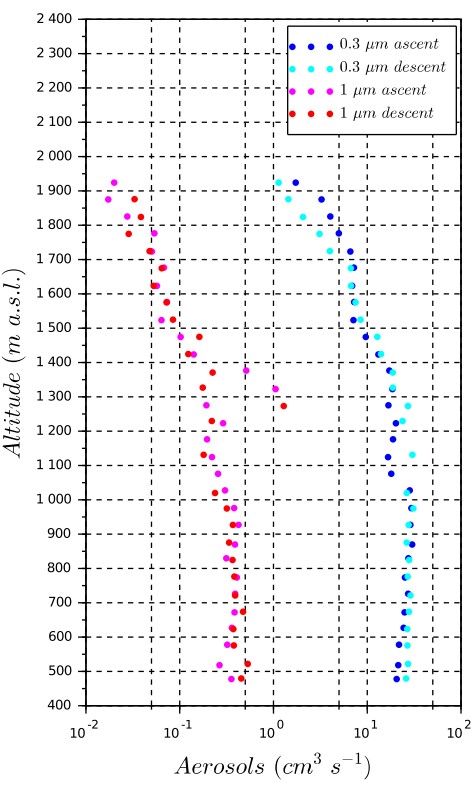

**Figure 10.** Vertical profiles of aerosol number concentration for a previous flight an hour earlier (Flight 65) greater 0.3 $\mu$m and greater 1 $\mu$m.



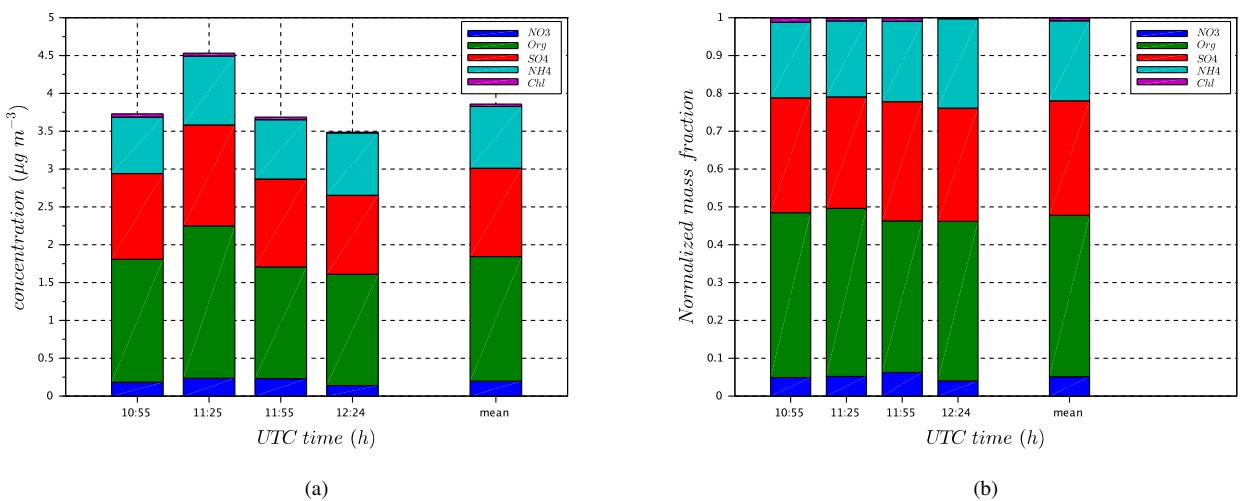

(a)  (b)

**Figure 11.** a) Concentration of aerosol chemical composition during the flight (Aerosol chemical speciation monitor measurement). b) Normalized mass fraction of aerosol chemical composition during the flight.

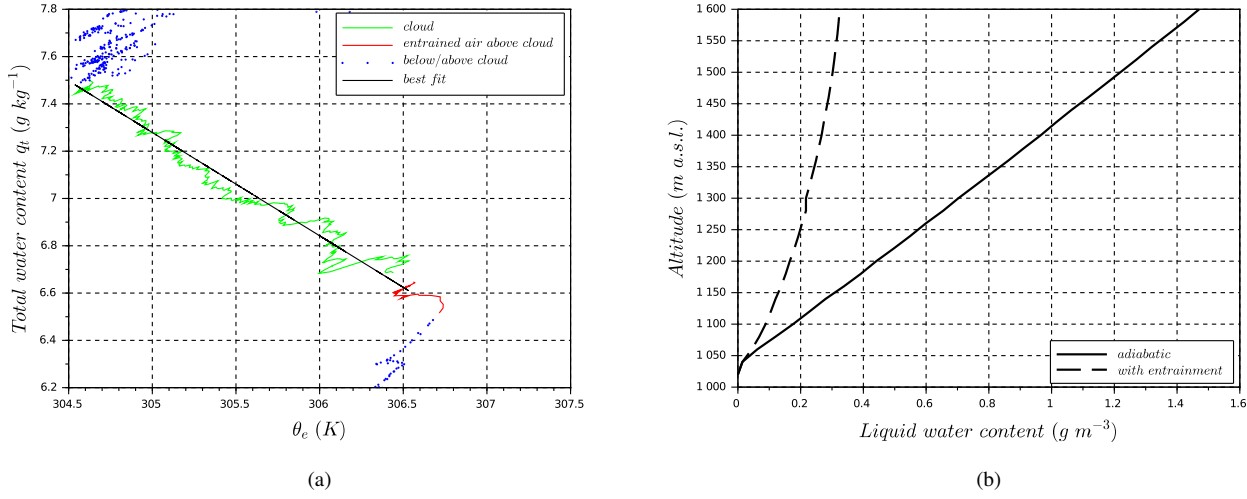

(a)  (b)

**Figure 12.** a) Total water content $q_t$ and equivalent potential temperature $\theta_e$ identify mixing between cloud air and entrained air. b) Liquid water content in cloud calculated for an adiabatic profile and when the entrainment is considered.



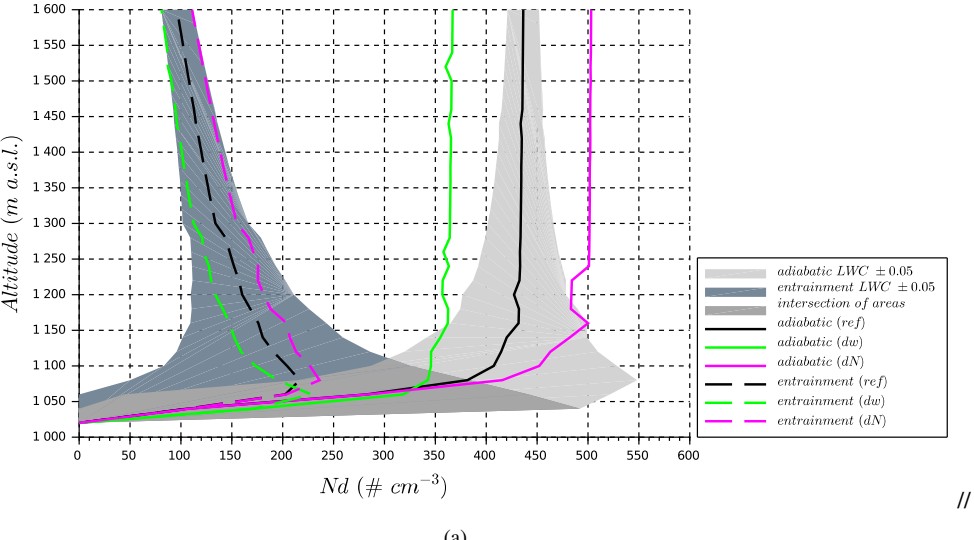

(a)

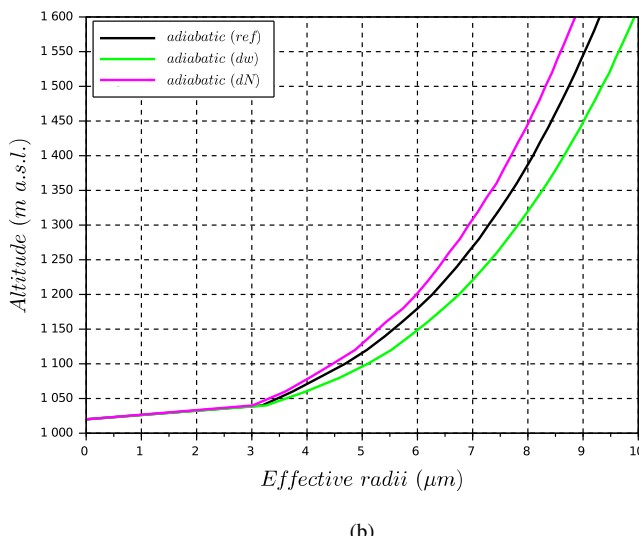

(b)

**Figure 13.** a) Simulated cloud droplet number function of the cloud height, for the adiabatic and the entrainment cases, with variation of updraft velocity ($dw$) and particle number ($dN$). b) Cloud effective radii for the adiabatic case with variation of updraft velocity ($dw$) and particle number ($dN$).



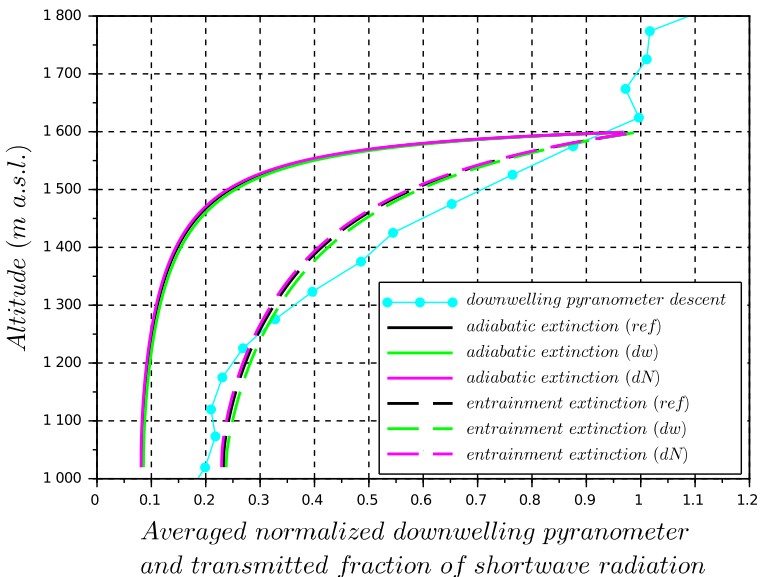

**Figure 14.** Optical cloud profile, comparison of pyranometer profiles and model simulations for the normalized transmission, with adiabatic and entrainment processes. Cloud top is at 1600 m a.s.l.. Cloud base is at 1020 m a.s.l.

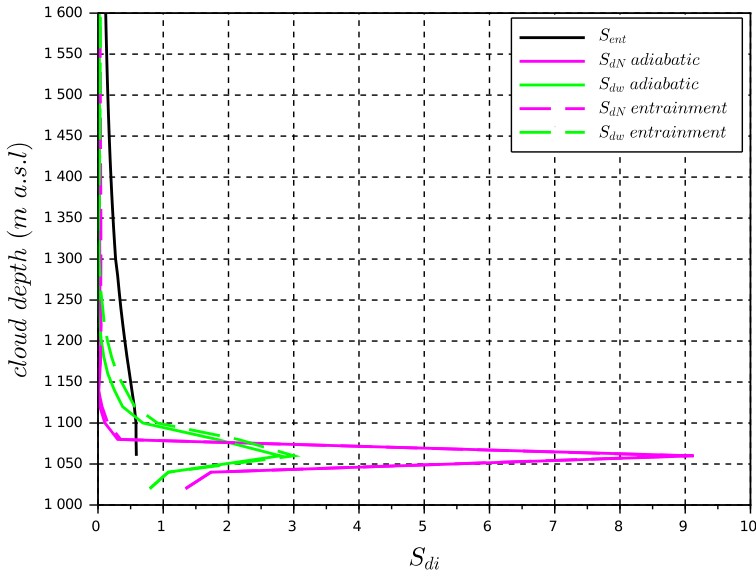

**Figure 15.** Sensitivity of albedo function of cloud depth, $S_{di}$ is defined in Eq.10.