# Peer review of "Aerosol-Cloud Closure Study on Cloud Optical Properties using Remotely Piloted Aircraft Measurements during a BACCHUS Field Campaign in Cyprus"

_Atmospheric Chemistry and Physics, 2019_

## Referee Comment (RC1) · James Hudson (Referee) · 8 May 2019

Entrainment is certainly important for clouds and for cloud effects. But the statement in the last sentence that entrainment reduces the impact of cloud radiative forcing is patently untrue and needs to be deleted. Entrainment is certainly a complicating factor for evaluating the indirect aerosol effect (IAE), but it is an unlikely mitigator of IAE. It mainly makes IAE more difficult to evaluate. This manuscript makes this point but it in no way shows that entrainment reduces IAE. Alterations of entrainment are not part of any IAE scenarios or any climate change scenarios. Just because there might be

more variations of optical properties due to entrainment variations than there might be due to certain aerosol variations does not mean that IAE is in any way mitigated by entrainment. The entrainment variations will occur with or without IAE. It does not take any knowledge of cloud physics to see the logical flaw of this contention. IAE will affect adiabatic clouds and it will affect subadiabatic clouds. We certainly need to know about and understand entrainment but not just for IAE. The factor of 2 greater N and the factor of 2 lower W are completely arbitrary and have no relationship with IAE. Moreover, W is not purported to change with IAE or any other possible climate alterations. The cloud microphysical analysis based on inhomogeneous mixing and Fig. 13a are at odds with the 100 nm Hoppel minimum with 400 cm-3. If there are really only 200 cm-3 droplets, then the Hoppel minimum should occur where there are 200 cm-3. With inhomogeneous mixing there is complete indiscriminate evaporation of some droplets (Baker et al., 1980; Telford & Wagner, 1981). The raison d'etre for inhomogeneous (or entity) mixing was to reduce droplet concentration so that sizes would be large enough to initiate coalescence even with high CCN concentrations. Cloud droplets are evaporated with no relationship to their size or to the CCN that they were grown upon. Therefore, inhomogeneous mixing disrupts the relationship between CCN and cloud droplets. Thus, inhomogeneous mixing itself mitigates IAE because it severs the link between CCN and cloud droplet concentration and thus all cloud microphysics. Since surface aerosol measurements are a basic component of this analysis this presents a large disruption to this analysis. On the other hand, I have several papers that show that CCN remain related to droplet concentrations in spite of entrainment; these include those cited in this manuscript and Hudson et al. (2018). As stated in these papers homogeneous rather than inhomogeneous mixing is thus indicated. The analysis presented here is too farfetched to be considered closure. Using a model as a proxy for Nc is unreasonable, especially when inhomogeneous mixing is assumed. Droplet spectral width seems to be ignored. It too can vary and have consequences. Some clouds that are less susceptible to IAE because they already altered by IAE. This is just the law of diminishing returns, which does not even require cloud physics knowledge. Pollution

will have less effect on clouds that are already somewhat polluted. Vertical velocity is a redundant term. Velocity is speed with direction. Vertical is a direction. Vertical wind is not redundant. Some abbreviate cloud droplet concentrations Nc others use Nd. Nd should be reserved for drizzle drop concentration. Hudson et al. (2012) showed that mixing of cloud parcels with various W can help explain droplet spectral broadening. There is no entrainment at cloud base. Cloud albedo does not approach 1.

P4. L 2. 100 nm in Fig. 3a looks more like a shoulder than a minimum. L12. I do not know what you are referring to in Hudson et al. (2015). L20. Delete long. L26. What size range for the Q-ACSM? P5. L12. Insert critical before supersaturation. P5. L17. This is not the critical diameter (dc). This would be the diameter of a droplet at the peak of the Kohler curve. It is the largest unactivated haze droplet or the minimum size of an activated droplet. For 0.24% Sc this is 0.58 $\mu$m (580 nm). This is not meant here. Apparently, this is the dry diameter that corresponds to 0.24% Sc. But this would depend on particle composition. For NaCl this is 56 nm, for amon. sul. 70 nm. This is not consistent with the later kappa discussion that suggests 0.3, for a mixture of ammonium sulfate and organics. This does make sense for 0.24% Sc and 100 nm. Later the term critical dry diameter is used, and this is apparently what is meant here but where does 94.5 nm originate? P6. L31. I do not see how this decrease can be observed within cloud where it is supersaturated, and RH cannot be measured. L34. The air mass is not saturated only the cloud is saturated. P7. L20-1. That may be true for the simulations, but it may not be true for the real clouds being investigated. P12. L15. This will depend very much on the width of the droplet spectra. L27. Lower W causes lower S and lower Nc but it does not directly cause larger droplet sizes. This might be the case if LWC remains constant. L34. This would depend on the initial values. What are they? P14. L16. This depends on the initial Nc and droplet size distribution.

Baker, M.B., Corbin, R.G., & Latham, J. (1980). The influence of entrainment on the evolution of cloud droplet spectra: I. A model of inhomogeneous mixing. Quarterly Journal of the Royal Meteorological Society, 108, 871-898. Hudson, J.G., S. Noble and V. Jha, 2012: Cloud droplet spectral width relationship to CCN spectra and vertical velocity. Journal of Geophysical Research: Atmospheres, 117, D11211, doi:10.1029/2012JD017546. Hudson, J.G., S. Noble, and S. Tabor, 2018: CCN spectral shape and stratus cloud and drizzle microphysics. Journal of Geophysical Research: Atmospheres, 123, 9635-9651. http://doi.org/10.1029/2017JD027865 Telford J.W., and Wagner, P.B. (1981). Observation of condensation growth determined by entity type mixing. Pure and Applied Geophysics, 119, 934-965.

---

## Referee Comment (RC2) · Anonymous Referee #2 · 18 Jun 2019

The manuscript by Calmer et al. shows results from an aerosol-cloud closure study based on ground-based and airbourne measurements complemented by a numerical cloud parcel model. The basic methods used in the investigation are sound. The numerical modelling framework seems rather simplistic, but is probably adequate to characterize the aerosol-cloud closure in the stratiform cloud case. Moreover, it is nice to see that the effect of entrainment on the aerosol-cloud coupling is considered, which comprises some of the more interesting aspects of the manuscript.

I'd like to ask the Authors to improve the presentation quality of the model description

and some of the results, which seem very confusing and hard to follow at times. Please see the specific comments for details.

In addition, I think the contribution by the simple sensitivity tests to the outcome of the manuscript is rather weak, since it is very clear from the basic analysis (as well as from other sources) that neglecting such a central physical process as entrainment will not yield a good aerosol-cloud closure in most cases. It would be more interesting to try characterize e.g. the possible feedback effects between the aerosol-cloud microphysical perturbations and entrainment. Other than the sensitivity tests, the manuscript does not touch on the subject of aerosol indirect effects specifically, and the sensitivity tests presented are not really enough to make any conclusions about AIE. I suppose the currently used model setup is not able to adjust entrainment in response to changes in the cloud radiative properties caused by changing droplet number etc.?

Specific comments:

Section 3: The model description is not adequate: is it actually a 1-D column model, or a (0-D) parcel model which you lift along a vertical trajectory? Where and how do you set the model top, do you limit it exactly at cloud top height? How long timestep do you use? Please be more precise and put some more effort to details.

Page 7, line 18: Aerosols are given in fixed bins, but what is the size range and spacing of the bins? The sentence on line 26 talks about the aerosol particles again, while the previous sentence is about liquid water in "moving section representation"? Please describe one issue at a time (i.e. aerosol bin characteristics, cloud droplet bin characteristics, the coupling between aerosol and cloud etc.) to make it easier to read and, again, try to make it consistent and precise.

Page 9, line 6:

Page 11, line 15: In reality, cloud albedo remains somewhat smaller than unity. This assumption does not seem legitimate.

Section 4.3: Please declare the notation for the experiments (dw and dN) already in the beginning of the section to make it easier to read.

Section 4.3: Please check and revise the presentation of the results. First you state that halving the updraft velocity decreases the fraction of transmitted shortwave radiation. Do you mean increases, since this contradicts the change shown in Fig 14 as well as the general expectation, or do you change the point of reference from the control experiment to the dw experiment? In the latter case, please don't do that. Also, the reported total range of change seen in the transmitted fraction (dN effect as the low limit, dw effect as the high) then adds to this confusion.

Page 12, lines 28-29: "The lower cloud droplet number and smaller effective radii results in lower albedo" – do you mean larger effective radii? Does all this still refer to the dw experiment? Please clarify.

Page 14, line 18: Combining the range of variation in the cloud radiative properties from the dw and dN sensitivity experiments feels a bit awkward, since the perturbations used in the two quantities and their magnitudes have nothing to do with each other whatsoever. Please consider keeping the results from these sensitivity experiments separate.

Page 14, line 25: "... highlight notably that entrainment processes reduce the impact of cloud radiative forcing." Please reconsider this statement. This generalization is not the same thing as having biased model results because of neglecting this essential physical process. Instead, referring to the general comment in the beginning of this review, entrainment can react e.g. to aerosol induced changes in the cloud and create a feedback loop, but this is not investigated in the manuscript.

Technical comments

Figure 10: Please rephrase the caption; "number of particles greater than 0.3 um" etc.

Figure 11: Please rephrase the caption; "mass concentration and chemical composition of aerosol during the flight ..." etc.

Figure 13: Please explain the shaded areas briefly in the caption.

Figure 14: The x-axis label is pretty confusing, try to simplify.

Page 3 lines 21-22 unclear sentence

Are the model code and data publicly available?

---

## Author Comment (AC1) · 15 Aug 2019

The comment was uploaded in the form of a supplement:
https://www.atmos-chem-phys-discuss.net/acp-2019-8/acp-2019-8-AC1-supplement.pdf

---

## Author Response (AR1)

**Interactive comment on "Aerosol-Cloud Closure Study on Cloud Optical Properties using Remotely Piloted Aircraft Measurements during a BACCHUS Field Campaign in Cyprus"**

James Hudson (Referee)

hudson@dri.edu

We thank the reviewer for his critical comments and corrections, which strengthen the quality of this manuscript.

Comments from the reviewer appear in italic, response from the authors follows.

*Entrainment is certainly important for clouds and for cloud effects. But the statement in the last sentence that entrainment reduces the impact of cloud radiative forcing is patently untrue and needs to be deleted.*

Response: This sentence was imprecise and not intended to suggest a link between entrainment and the impact of cloud forcing. It has been rephrased to avoid any confusion.

In the manuscript current text : "The case studies in Cyprus (this study) and at Mace Head illustrate the significance of the entrainment processes in determining cloud optical properties in two different environments, and highlight notably that entrainment processes reduce the impact of cloud radiative forcing."

Updated text : "The case studies in Cyprus (this study) and at Mace Head illustrate the significance of the entrainment processes in determining cloud optical properties in two different environments. Entrainment mixing decreases the water content in the cloud relative to an adiabatic profile, therefore not taking into account entrainement leads to a significant overestimation of cloud radiative forcing."

*Entrainment is certainly a complicating factor for evaluating the indirect aerosol effect (IAE), but it is an unlikely mitigator of IAE. It mainly makes IAE more difficult to evaluate. This manuscript makes this point but it in no way shows that entrainment reduces IAE. Alterations of entrainment are not part of any IAE scenarios or any climate change scenarios. Just because there might be more variations of optical properties due to entrainment variations than there might be due to certain aerosol variations does not mean that IAE is in any way mitigated by entrainment. The entrainment variations will occur with or without IAE. It does not take any knowledge of cloud physics to see the logical flaw of this contention. IAE will affect adiabatic clouds and it will affect subadiabatic clouds. We certainly need to know about and understand entrainment but not just for IAE.*

Response: We clarify the entrainment is not used for evaluating the indirect aerosol effect, and, yes, we do agree that variability in entrainment does make IAE more difficult to evaluate, which is one of the points that we attempt to highlight in this study. Nowhere in this manuscript do we intend to suggest that IAE and entrainment are related.

*The factor of 2 greater N and the factor of 2 lower W are completely arbitrary and have no relationship with IAE.*

Response: Yes, the factor of two changes are somewhat arbitrary, but only to show relative sensitivity. We follow similar methods where factor of two changes in aerosol concentrations are used in model sensitivity studies [e.g., Pringle et al. (2009); Moore et al. (2013)]. We are not suggesting $w$ is purported to change with IAE or any other possible climate alterations.

*The cloud microphysical analysis based on inhomogeneous mixing and Fig.13a are at odds with the 100 nm Hoppel minimum with 400 $cm^{-3}$. If there are really only 200 $cm^{-3}$ droplets, then the Hoppel minimum should occur where there are 200 $cm^{-3}$.*

Response: Since cloud base is close to adiabatic, nearly 400 $cm^{-3}$ will activate. Even as many of the particles subsequently evaporate, there will be a shift in the aerosol number size distribution towards the accumulation mode. We originally just used the time for the individual RPA flight, however, the ground-based observations were several kilometers away and are not exactly coincident to the specific airborne observations. Therefore, Fig.3a has been updated to show Hoppel minimum related to a longer time period (from Fig.2) that represents the aerosol-cloud interactions over the region of the RPA flights.

*With inhomogeneous mixing there is complete indiscriminate evaporation of some droplets (Baker et al., 1980; Telford & Wagner, 1981). The raison d'être for inhomogeneous (or entity) mixing was to reduce droplet concentration so that sizes would be large enough to initiate coalescence even with high CCN concentrations. Cloud droplets are evaporated with no relationship to their size or to the CCN that they were grown upon. Therefore, inhomogeneous mixing disrupts the relationship between CCN and cloud droplets. Thus, inhomogeneous mixing itself mitigates IAE because it severs the link between CCN and cloud droplet concentration and thus all cloud microphysics. Since surface aerosol measurements are a basic component of this analysis this presents a large disruption to this analysis. On the other hand, I have several papers that show that CCN remain related to droplet concentrations in spite of entrainment; these include those cited in this manuscript and Hudson et al. (2018). As stated in these papers homogeneous rather than inhomogeneous mixing is thus indicated.*

Response: While we agree homogeneous entrainment can occur in clouds, a number of previous studies have shown that stratocumulus cloud-top entrainment specifically results in inhomogeneous mixing (Brenguier et al. (2011); Burnet et al. (2007); Yum et al. (2015); Painemal et al. (2011)). In a stratocumulus study, Painemal et al. (2011) showed the effective radius does not decrease despite a reduction in LWC. While inhomogeneous entrainment may weaken direct correlations between CCN and CDNC, the relationship is not expected to be eliminated. Similarly other dynamic processes controlling supersaturation, droplet nucleation, and droplet growth within clouds lead to non linear relationship between CCN and CDNC although correlations often still exist between CDNC and CCN (ex. Hudson et al. (2010), Figure 1).

*The analysis presented here is too farfetched to be considered closure. Using a model as a proxy for Nc is unreasonable, especially when inhomogeneous mixing is assumed.*

Response: We disagree, many closure experiments using a similar model have already been published in the literature (Conant et al. (2004); Peng et al. (2005); Fountoukis et al. (2007); Sanchez et al. (2017)).

*Droplet spectral width seems to be ignored. It too can vary and have consequences. Some clouds that are less susceptible to IAE because they already altered by IAE. This is just the law of diminishing returns, which does not even require cloud physics knowledge.*

Response: Droplet spectral width is not ignored. Equation 7, which is used to calculate the integrated cloud droplet extinction, is based on the droplet size distribution from the parcel model. The droplet extinction is used to determine the integrated cloud optical properties for comparison of simulations with and without entrainment. We propose an analysis of the droplet width sensitivity in a later comment.

*Pollution will have less effect on clouds that are already somewhat polluted.*

Response: Indeed, this point has been stated in Section 4.3.

In the manuscript current text : "Clouds formed in cleaner environments are likely to be of higher susceptibility compared to clouds in polluted areas, which illustrate the link between pollution and cloud albedo proposed by Twomey (1977)."

*Vertical velocity is a redundant term. Velocity is speed with direction. Vertical is a direction. Vertical wind is not redundant.*

Response: We agree, and it has been corrected through the paper to state 'vertical wind' or 'updraft'.

*Some abbreviate cloud droplet concentrations Nc others use Nd. Nd should be reserved for drizzle drop concentration.*

Response: We agree that different abbreviates are have been used throughout the literature, e.g., in Conant et al. (2004), cloud droplet number concentration is $N_D$. In Fountoukis et al. (2007), it is $N_d$, in Sanchez et al. (2017), it is CDNC. In this study, it had been chosen as $N_d$ to be consistent with the other closure studies with airborne measurements as drizzle drop concentrations are not discussed. However, to avoid any confusion, $N_d$ has been replaced by CDNC in the manuscript.

*Hudson et al. (2012) showed that mixing of cloud parcels with various W can help explain droplet spectral broadening.*

Response: A weighted ensemble of updrafts were applied to the ACPM to account for spectral broadening due to variations in $w$. The following text was included to address this point.

In the manuscript current text : "In the literature, either a characteristic updraft velocity or an updraft distribution is used in ACPM. Conant et al. (2004), Hsieh et al. (2009), and Sanchez et al. (2017) have shown that a distribution of updraft velocities and a weighted distribution of in-cloud supersaturations better reproduce cloud microphysical properties than a single-updraft approximation. Consequently, positive vertical wind velocity distribution near cloud base are used as model input (updraft velocities from 0.1 to 4 m s$^{-1}$ in Fig.5)."

Updated text: "In the literature, either a characteristic updraft or a distribution of updrafts is used in ACPM. Conant et al. (2004), Hsieh et al. (2009), Hudson et al. (2012), and Sanchez et al. (2017) have shown that accounting for the distribution of updrafts better reproduces cloud microphysical properties, such as the droplet spectral width, than a single-updraft approximation. Consequently, a weighted distribution of the positive vertical winds near cloud base is used as model input (updrafts from 0.1 to 4 m s$^{-1}$ shown in Fig.5), resulting in a broader cloud droplet distribution than when using a single updraft."

*There is no entrainment at cloud base.*

Response: Indeed, there is no entrainment at cloud base, only at cloud top; however, the dry entrained air (from above the cloud) is mixed downward throughout the cloud and the decrease in total water content affects the entire cloud. This process has been explained in Wang et al. (2009), and the following text has been updated :

In the manuscript current text : "To apply the cloud-top mixing, a fraction of air at cloud base and a fraction of air above cloud top are mixed, conserving the total water content and the equivalent potential temperature (Sanchez et al., 2017)."

Updated text : "To apply the cloud-top mixing, which corresponds to the dry entrained air from the cloud-top incorporated downward throughout the cloud, a fraction of air at cloud base and a fraction of air above cloud top are mixed, conserving the total water content and the equivalent potential temperature (Sanchez et al. (2017); Wang et al. (2009))."

*Cloud albedo does not approach 1.*

Response: the text has been updated to avoid any confusions.

In the manuscript current text : "As the cloud thickens, the albedo approaches unity meaning that all incoming solar irradiance is reflected back to space (Fig.14)."

Updated text : "As the cloud thickens, the albedo increases (but remains less than 1) meaning that more incoming solar irradiance is reflected back to space (Fig.14)."

*P4. L2. 100 nm in Fig.3a looks more like a shoulder than a minimum.*

Response: As mentioned in an earlier comment, as cloud base is adiabatic and nearly 400 droplets cm$^{-3}$ activate (and evaporate), then there were a shift to the accumulation mode as we see. Note that aerosol particles go through multiple cloud cycles before wet deposition, such that most CCN will have gone through multiple clouds before sampling at the ground. Fig.3a has been redrawn with a longer time period averaged to highlight that the break (related to the Hoppel minimum) at 100 nm is related to aerosol cloud-interactions. As mentioned previously, this longer time average also better represents the conditions over the period of the experiment.

[Figure]

(a)                                        (b)

Figure 1: a) Particle size distribution showing combined data from the SMPS (blue), the ground-based OPC (magenta) and the RPA-OPC (green). The red line indicates the Hoppel minimum diameter. b)Contour plot showing time series of SMPS data 2015/04/01. The black rectangle represents the selected data for the analysis (8 hours) and the magenta lines delimit the flight (Flight 67). Local pollution, which is not representative of the regional aerosol, has been removed (white).

*L12. I do not know what you are referring to in Hudson et al. (2015).*

Response: We thank the reviewer for catching this mistake. The reference has been updated.

In the manuscript current text: "On the time scale of hours, the inactivated CCN, or interstitial aerosol, do not change critical size or supersaturation SS (Hudson et al., 2015)."

Updated text: "On the time scale of hours, the inactivated CCN, or interstitial aerosol, do not change critical size or supersaturation SS (Hoppel et al., 1996)."

*L20. Delete long.*

Response: 'long' has been deleted.

In the manuscript current text : "the wingspan is 1.5 m long."

Updated text: "the wingspan is 1.5 m."

*L26.What size range for the Q-ACSM?*

Response : The size range for the Q-ACSM is 40 nm to 1 $\mu$m diameter.

In the manuscript current text : "An aerosol chemical speciation monitor (Q-ACSM, Aerodyne Research Inc) provides the chemical composition of non-refractory submicron aerosol particles."

Updated text: "An aerosol chemical speciation monitor (Q-ACSM, Aerodyne Research Inc.) provides the chemical composition of non-refractory submicron aerosol particles with a range from 40 nm to 1 $\mu$m diameter."

*P5. L12. Insert critical before supersaturation.*

Response : 'critical' has been inserted.

In the manuscript current text : "On the time scale of hours, the inactivated CCN, or interstitial aerosol, do not change size or supersaturation SS."

Updated text : "On the time scale of hours, the inactivated CCN, or interstitial aerosol, do not change size or critical supersaturation SS."

*P5.L17. This is not the critical diameter (dc). This would be the diameter of a droplet at the peak of the Kohler curve. It is the largest unactivated haze droplet or the minimum size of an activated droplet. For 0.24 % Sc this is 0.58 µm (580 nm). This is not meant here. Apparently, this is the dry diameter that corresponds to 0.24 % Sc. But this would depend on particle composition. For NaCl this is 56 nm, for amon. sul. 70 nm. This is not consistent with the later kappa discussion that suggests 0.3, for a mixture of ammonium sulfate and organics. This does make sense for 0.24 % Sc and 100 nm. Later the term critical dry diameter is used,*

*and this is apparently what is meant here but where does 94.5 nm originate?*

Response: $N_{CCN}$ observations at 0.24 % give a cumulative number of particles of 420 cm$^{-3}$, which corresponds to the diameter of 94.5 nm on the Fig.3b (solid cyan line) by integrating the aerosol number size distribution.

In the manuscript current text : "Similarly, based on the CCN measurements at the ground-station, the CCN concentration at 0.24 % SS corresponds to 420 cm$^{-3}$. These results imply that a characteristic in-cloud supersaturation is close to 0.24 % SS. The critical diameter at 0.24 % SS is 94.5 nm."

Updated text : "Similarly, based on the CCN measurements at the ground-station, the CCN concentration at 0.24 % SS is 420 cm$^{-3}$, which corresponds to a dry critical diameter of 94.5 nm (Fig.3b) and is similar to the Hoppel minimum in Fig.13a. These results suggest that a characteristic in-cloud supersaturation is close to 0.24 % SS."

*P6. L31. I do not see how this decrease can be observed within cloud where it is supersaturated, and RH cannot be measured.*

Response: The text has been updated to avoid any confusion.

In the manuscript current text : "The relative humidity increases from 75 % at the ground to 100 % at the cloud base (1020 m a.s.l.), and then decreases again closer to the cloud top (Fig.9b)."

Updated text : "The relative humidity increases from 75 % at the ground to 100 % at the cloud base (1020 m a.s.l.), and then decreases again at cloud top (Fig.9b)."

*L34. The air mass is not saturated only the cloud is saturated.*

Response: The sentence has been updated.

In the manuscript current text : "the measured values have been scaled such that RH measurements are 100 % in a cloud (the air mass is considered saturated)."

Updated text : "The measured values have been scaled such that the air inside the cloud is saturated (i.e., RH is 100 %)."

*P7. L20-1. That may be true for the simulations, but it may not be true for the real clouds being investigated.*

Response: The case studied in the present manuscript is a non-precipitating cloud, and assumptions of negligible impacts of droplet collision, coalescence and drizzle rates are based on Feingold et al. (2013).

In the manuscript current text : "Droplet collision, coalescence and drizzle rates are negligible for the simulated values of liquid water content and cloud droplet number concentration in the

present case (i.e. droplet diameter D p < 20 $\mu$m).''

Updated text : ''The case study focuses on a non-precipitating cloud (i.e., droplet diameter < 20 $\mu$m); therefore, droplet collision, coalescence and drizzle rates are negligible for the simulated values of liquid water content and cloud droplet number concentration.''

*P12. L15. This will depend very much on the width of the droplet spectra.*

Response: As mentioned in a previous comment, the impact of the droplet spectral width on the transmitted fraction of shortwave radiation has been investigated. Cloud droplet distributions from the parcel model have been represented by lognormal distributions and the width of the distribution ($\sigma$) was changed by a factor of two, while keeping total number of droplets and the LWC equal to the reference case. The difference in the fraction of transmitted radiation is 0.002 for a factor of two change in the droplet spectral width. This difference is nearly the same as the changes in transmitted fraction based on a factor of two changes in aerosol (0.002) and updraft (0.003) in the adiabatic case, which is still much smaller than the difference between adiabatic and entrainment cases.

[Figure]

Figure 2: Droplet size distribution at 1300 m a.s.l., the black solid line corresponds to CDNC calculated from the parcel model (reference case). The blue and red lines are lognormal distributions of the corresponding output distribution with equal CDNC and LWC, only the standard deviation $\sigma$ of the distributions varies.

[Figure]

Figure 3: Calculated extinction as a function of altitude with different cloud droplet number spectral widths, with $\sigma$ and $\sigma/2$.

[Figure]

Figure 4: Transmitted fraction of shortwave radiation as a function of altitude for the adiabatic case compared to the impact of factor of two changes in droplet spectral width on the transmitted fraction of shortwave radiation.

[Figure]

Figure 5: Zoom at cloud base on transmitted fraction of shortwave radiation for the reference adiabatic case compared to a factor of two change in updraft (*dw*), aerosol number (*dN*), and σ for the associated lognormal distributions (CDNC and LWC remain equal to the reference case).

Updated text : "The impact of a change in the droplet spectral width has also been studied, using two lognormal droplet distributions with a factor of two variation in the standard deviation (σ) while the total number of droplets and liquid water content remain the same as the reference case.
A factor of two change in the spectral width showed an even smaller difference of 0.002 in the fraction of transmitted shortwave radiation at cloud base."

*L27. Lower W causes lower S and lower Nc but it does not directly cause larger droplet sizes. This might be the case if LWC remains constant.*

Response: We do not see any reason why the LWC would not remain constant under these conditions. However, the sentence has been changed to clarify this point.

In the manuscript current text : "The lower vertical wind velocities also results in lower cloud droplet number concentrations and larger effective radii owing to lower in-cloud supersaturations (Fig.13)."

Updated text : "The smaller updrafts also result in lower cloud droplet number concentrations owing to lower in-cloud supersaturations (Fig.13)."

*L34. This would depend on the initial values. What are they?*

Response: We agree, changes in the fraction of transmitted shortwave radiation depend on the initial values, which are summarized in Table 1 and 3, based on Figures 3 and 5.

*P14. L16. This depends on the initial Nc and droplet size distribution.*

Response: We agree and updated the manuscript to clarify this point.

In the manuscript current text : "A doubling of $N$ increases the maximum cloud droplet number by 50 cm$^{-3}$, whereas a reduction in $w$ decreases the maximum cloud droplet number by 70 cm$^{-3}$. The impact on cloud effective radius is relatively small, less than $\pm$ 1 $\mu$m changes in the radius ($<$ 7 % in relative changes)."

Updated text : "In addition to comparing ACPM results between entrainment and adiabatic cases, a sensitivity analysis presented here explores the impact of a change in aerosol particle number concentrations ($dN$) as well as changes in the updraft distribution ($dw$) on the cloud optical properties (Pringle et al. (2009); Moore et al. (2013)). Profiles of the cloud droplet number and effective radii (Fig.13) and cloud optical properties (Fig.14) are also simulated with the inputs of aerosol number concentration multiplied by two ($dN=2N$) and the updraft distribution divided by two ($dw=w/2$). Increasing the aerosol concentrations by a factor of two results in an aerosol concentration of $\sim$ 2400 cm$^{-3}$ representing more polluted conditions. Such an increase in aerosol/CCN concentrations also increases cloud droplet number concentration (Fig.13a), decreases the effective radii (Fig.13b), and presents a cloud with a higher albedo. In addition, halving the updraft distribution results in a distribution with maximum vertical wind near 2 m s$^{-1}$, which also happen to be similar to the updrafts observed in marine stratocumulus cloud layers over Mace Head Research Station, Ireland (Calmer et al., 2018). In this case study of $dw$, the lower updrafts also result in lower cloud droplet number concentrations with larger effective radii owing to lower in-cloud supersaturations (Fig.13). The lower cloud droplet number and larger effective radii results in lower albedo of the cloud layer and an increase of the fraction of transmitted shortwave radiation (Fig.14). In the adiabatic case, a decrease of 16 % in cloud droplet number is observed when the updraft distribution is divided by two ($dw$); and an increase of 11 % of droplet number occurs when the number of dry particles is multiplied by two ($dN$). The impact of a change in the droplet spectral width has also been studied, using two lognormal droplet distributions with a factor of two variation in the standard deviation ($\sigma$) while the total number of droplets and liquid water content remain the same as the reference case. Factor of two changes in updraft distribution causes the fraction of transmitted shortwave radiation to increase by 0.003 in the adiabatic case, and 0.005 in the entrainment case, corresponding to an decrease in albedo. Likewise, a factor of two increase in aerosol size distribution leads to a -0.002 (adiabatic case) and -0.004 (entrainment case) decrease in the fraction of transmitted shortwave radiation through the cloud (corresponding to similar net increase in cloud albedo, Fig.14). A factor of two change in the droplet spectral width showed an even smaller difference of 0.002 in the fraction of transmitted shortwave radiation at cloud base. To summarize, factor of two variations of $N$, $w$, and droplet spectral width correspond to a change within $\pm$ 0.005 in the transmitted shortwave radiation (and albedo) compared to the reference case. Yet, the change in the fraction of transmitted shortwave radiation between adiabatic and entrainment cases is 0.15, corresponding to a factor of thirty change in cloud albedo compared to changes in droplet number, updraft, and spectral width. The impact of entrainment on cloud optical properties has long been known (Boers et al., 1994), and this study only emphasizes its impact relative to aerosol indirect effect, changes in vertical motion, and cloud droplet spectral width. Extending this analysis further suggests that the sensitivity of cloud optical properties related to entrainment variability also needs to be constrained in order to improve climate models."

Response: We acknowledge that the cloud parcel model is relatively simplistic, but we specifically choose to do so to conduct the closure experiment, directly compare with in-situ measurements, and quantify the impact of entrainment relative to other microphysical processes.

*I'd like to ask the Authors to improve the presentation quality of the model description and some of the results, which seem very confusing and hard to follow at times. Please see the specific comments for details.*

Response: We improved the presentation quality by rephrasing the model description. The updated text is shown under the specific comments from the reviewer (Section 3).

*In addition, I think the contribution by the simple sensitivity tests to the outcome of the manuscript is rather weak, since it is very clear from the basic analysis (as well as from other sources) that neglecting such a central physical process as entrainment will not yield a good aerosol-cloud closure in most cases. It would be more interesting to try characterize e.g. the possible feedback effects between the aerosol-cloud microphysical perturbations and entrainment.*

Response: We are well aware from the literature that entrainment constitutes a central physical processes; however, we are not aware of other studies that have directly quantified the impact of entrainment on cloud optical properties relative to changes in aerosol and vertical winds. In the parcel model, aerosol and vertical winds determine the cloud properties from the model; however, only for an adiabatic parcel. The impact of entrainment is determined by comparison with in situ observations and one of the short coming with the parcel model is that it is not possible to study feedback effects.

*Other than the sensitivity tests, the manuscript does not touch on the subject of aerosol indirect effects specifically, and the sensitivity tests presented are not really enough to make any conclusions about AIE.*

Response: The aerosol indirect effect (AIE) is only discussed to study relative impact of aerosol on cloud optical properties and is not the focus of the paper. The factor of two variations in aerosol and updraft are arbitrary and are designed to explore the sensitivity related to cloud optical properties as has been done in other simulations (Pringle et al. (2009); Moore et al. (2013)), and compare the sensitivity to the effect of entrainment. We have stated in the manuscript that our study focuses on a relatively polluted case and is inherently less sensitive (w.r.t. aerosols) than a case with lower aerosol concentrations.

*I suppose the currently used model setup is not able to adjust entrainment in response to changes in the cloud radiative properties caused by changing droplet number etc.?*

Response: No, the model set up is not able to adjust entrainment in response to changes in the cloud radiative properties caused by changing droplet number. The correction for entrainment is applied to the model output based on observed thermodynamic properties.

*Specific comments:*
*Section 3: The model description is not adequate: is it actually a 1-D column model, or a (0-D) parcel model which you lift along a vertical trajectory? Where and how do you set the model top, do you limit it exactly at cloud top height? How long timestep do you use? Please be more precise and put some more effort to details.*

Response: We specifically kept the model description succinct and cited Sanchez et al. (2017), Russell et al. (1999) and Russell et al. (1998) for a more detailed description of the model and methods. The model is a 0-D parcel model in which a parcel is lifted vertically at the observed distribution of updrafts. The model is limited to the cloud top height (as observed). The time step is 0.1 seconds to account for kinetic limitations on droplet growth (Chuang et al., 1997). To address the

concerns of the reviewer, we have added the specific details and highlighted them below in the relevant text.

In the manuscript : "The 0-D Aerosol-Cloud Parcel Model (ACPM) is based on Russell et al. (1998) and Russell et al. (1999), where the main equations explicitly described the processes of activation of aerosol particles and the condensation of water vapor on the resulting cloud droplets. The model is designed to be initialized from aircraft-based field observations. The ACPM lifts a parcel of air along a vertical trajectory limited by the observed cloud top height, at timesteps of 0.1 seconds, to account for kinetic limitations in droplet growth (Chuang et al., 1997). The input aerosol particle distribution is divided into 70 bins that are equally log spaced with a minimum bin edge size of 0.02 $\mu$m, and a maximum bin edge size of 3.0 $\mu$m. The ACPM uses a fixed sectional approach for distinct aerosol populations to calculate particle growth under supersaturated conditions (Russell et al., 1998). ~~Evaporation from the entrainment is parameterized and applied to the ACPM results. Deposition is also included but negligible for the study here. Droplet collision, coalescence and drizzle rates are negligible for the simulated values of liquid water content and cloud droplet number concentration in the present case (i.e. droplet diameter $D_p <$ 20 nm). The model is designed to be initialized from aircraft-based field observations, and~~ The model employs a dual-moment (number and mass) algorithm to calculate the particle growth."

*Page 7, line 18: Aerosols are given in fixed bins, but what is the size range and spacing of the bins? The sentence on line 26 talks about the aerosol particles again, while the previous sentence is about liquid water in "moving section representation"? Please describe one issue at a time (i.e. aerosol bin characteristics, cloud droplet bin charac- teristics, the coupling between aerosol and cloud etc.) to make it easier to read and, again, try to make it consistent and precise.*

Response: The size range and spacing of the bins has been stated in the text, but we agree the organization of this section made it hard to follow. The first paragraph of section 3.1 has been reorganized to more clearly discuss the model details and is included in response to the previous comment. The updated text to describe the size range and spacing is:

In the manuscript current text : "To simulate drops in the model, aerosol particles are divided into 70 bins that are log spaced. The minimum size is 0.02 $\mu$m, and the maximum size is 3.0 $\mu$m."

Updated text: "The input aerosol particle distribution is divided into 70 bins that are equally log spaced with a minimum bin edge size of 0.02 $\mu$m, and a maximum bin edge size of 3.0 $\mu$m."

*Page 11, line 15: In reality, cloud albedo remains somewhat smaller than unity. This assumption does not seem legitimate.*

Response: Both reviewers have pointed this out. The text has been updated to avoid any confusion.

In the manuscript current text : "As the cloud thickens, the albedo approaches unity meaning that all incoming solar irradiance is reflected back to space (Fig.14)."

Updated text : "As the cloud thickens, the albedo increases (but remains less than 1) meaning that more incoming solar irradiance is reflected back to space (Fig.14)."

*Section 4.3: Please declare the notation for the experiments (dw and dN) already in the beginning of the section to make it easier to read.*

Response: Section 4.3 has been updated to declare the notation at the beginning.

In the manuscript current text : "In addition to comparing ACPM results between entrainment and adiabatic cases, a sensitivity analysis presented here explores the impact of a change in aerosol particle number concentrations as well as changes in the vertical velocity distribution on the cloud optical properties."

Updated text : "In addition to comparing ACPM results between entrainment and adiabatic cases, a sensitivity analysis presented here explores the impact of a change in aerosol particle number concentrations ($dN$) as well as changes in the updraft distribution ($dw$) on the cloud optical properties."

*Section 4.3: Please check and revise the presentation of the results. First you state that halving the updraft velocity decreases the fraction of transmitted shortwave radiation. Do you mean increases, since this contradicts the change shown in Fig 14 as well as the general expectation, or do you change the point of reference from the control experiment to the dw experiment? In the latter case, please don't do that. Also, the reported total range of change seen in the transmitted fraction (dN effect as the low limit, dw effect as the high) then adds to this confusion.*

Response : We thank the reviewer for point out the inconsistency. Halving the updraft velocity indeed increases the fraction of transmitted shortwave radiation. We clarify the point of reference from the control experiment.

In the manuscript : "In addition to comparing ACPM results between entrainment and adiabatic cases, a sensitivity analysis presented here explores the impact of a change in aerosol particle number concentrations *(dN)* as well as changes in the  updraft distribution *(dw)* on the cloud optical properties (Pringle et al. (2009); Moore et al. (2013)). Profiles of the cloud droplet number and effective radii (Fig.13) and cloud optical properties (Fig.14) are also simulated with the inputs of aerosol number concentration multiplied by two (*dN=2N*) and the updraft  distribution divided by two (*dw=w/2*). Increasing the aerosol concentrations by a factor of two results in an aerosol concentration of $\sim$ 2400 cm$^{-3}$ representing more polluted conditions. Such an increase in aerosol/CCN concentrations also increases cloud droplet number concentration (Fig.13a), decreases the effective radii (Fig.13b), and presents a cloud with a higher albedo. In addition, halving the  updraft distribution results in a distribution with maximum vertical wind  near 2 m s$^{-1}$, which also happen to be similar to the  updrafts observed in marine stratocumulus cloud layers over Mace Head Research Station, Ireland (Calmer et al., 2018).  In this case study of *dw*, the lower updrafts also result in lower cloud droplet number concentrations  with larger effective radii owing to lower in-cloud supersaturations (Fig.13). The lower cloud droplet number and  larger effective radii results in lower albedo of the cloud layer and an increase of the fraction of transmitted shortwave radiation (Fig.14).  In the adiabatic case, a decrease of 16 % in cloud droplet number is observed when the updraft  distribution is divided by two (*dw*); and an increase of  11 % of droplet number occurs when the number of dry particles is multiplied by two (*dN*). The impact of a change in the droplet spectral width has also been studied, using two lognormal droplet distributions with a factor of two variation in the standard deviation ($\sigma$) while the total number of droplets and liquid water content remain the same as the reference case. Factor of two changes in updraft distribution causes the fraction of

transmitted shortwave radiation to  increase by 0.003 in the adiabatic case, and 0.005 in the entrainment case, corresponding to an decrease in albedo. Likewise, a factor of two increase in aerosol size distribution leads to a -0.002 (adiabatic case) and -0.004 (entrainment case)  decrease in the fraction of transmitted shortwave radiation through the cloud (corresponding to similar net increase in cloud albedo, Fig.14). A factor of two change in the droplet spectral width showed an even smaller difference of 0.002 in the fraction of transmitted shortwave radiation at cloud base. To summarize, a factor of two variations of $N$, $w$, and droplet spectral width correspond to a change  within $\pm$ 0.005 in transmitted shortwave radiation (and albedo) compared to the reference case. Yet, the change in the fraction of transmitted shortwave radiation between adiabatic and entrainment cases is 0.15, corresponding to a factor of thirty change in cloud albedo compared to changes in droplet number, updraft, and droplet spectral width. The impact of entrainment on cloud optical properties has long been known (Boers et al., 1994) , and this study only emphasizes its impact relative to aerosol indirect effect and changes in vertical motion. Extending this analysis further suggests that the sensitivity of cloud optical properties related to entrainment variability also needs to be constrained in order to improve climate models."

*Page 12, lines 28-29: "The lower cloud droplet number and smaller effective radii results in lower albedo" - do you mean larger effective radii? Does all this still refer to the dw experiment? Please clarify.*

Response: We thank the reviewer for catching this mistake, and we have updated the manuscript for the *dw* experiment leading to larger effective radii and lower cloud droplet number concentration.

In the manuscript current text : "The lower vertical wind velocities also results in lower cloud droplet number concentrations and larger effective radii owing to lower in-cloud supersaturations (Fig.13) . The lower cloud droplet number and smaller effective radii results in lower albedo of the cloud layer (Fig.14)."

Updated text : "In this case study of *dw*, the lower vertical wind velocities also result in lower cloud droplet number concentrations owing to lower in-cloud supersaturations and larger effective radii (Fig.13). The lower cloud droplet number and larger effective radii results in lower albedo of the cloud layer or an increase of the fraction of transmitted shortwave radiation (Fig.14)."

*Page 14, line 18: Combining the range of variation in the cloud radiative properties from the dw and dN sensitivity experiments feels a bit awkward, since the perturbations used in the two quantities and their magnitudes have nothing to do with each other whatsoever. Please consider keeping the results from these sensitivity experiments separate.*

Response: To avoid confusion, we have chosen to simplify the summary of the sensitivity experiments.

In the manuscript current text : "These changes in cloud droplet number concentrations by varying N and w lead to changes between -0.004 and 0.005 in the fraction of transmitted shortwave radiation. In comparison, the change in fraction of transmitted shortwave radiation and albedo related to entrainment is 0.15."

Updated text : "To summarize, factor of two variations of *N*, *w*, and droplet spectral width

correspond to a change within $\pm$ 0.005 in transmitted shortwave radiation (and albedo) compared to the reference case. Yet, the change in the fraction of transmitted shortwave radiation between adiabatic and entrainment cases is 0.15, corresponding to a factor of thirty change in cloud albedo compared to changes in droplet number, updraft, and droplet spectral width."

*Page 14, line 25: ". . . highlight notably that entrainment processes reduce the impact of cloud radiative forcing." Please reconsider this statement. This generalization is not the same thing as having biased model results because of neglecting this essential physical process. Instead, referring to the general comment in the beginning of this review, entrainment can react e.g. to aerosol induced changes in the cloud and create a feedback loop, but this is not investigated in the manuscript.*

Response : We have corrected our phrasing of the quoted text. While it is clear that neglecting entrainment will result in inaccurate results, past closure studies do neglect including entrainment to focus on closure of cloud droplet number concentrations (particularly near cloud base). The impact of entrainment was observed for example in Conant et al. (2004), and Meskhidze et al. (2005); however, the data was screened and only on the case studies close to adiabatic values were studied. Feedbacks causing changes in entrainment due to aerosol induced changes is not included in the manuscript because it is out of the scope of this paper. Though these feedback effects are important, such a study cannot be conducted with a 0-D model as the model does not account for entrainment. The unique contribution of this paper is to use in-situ ground-based and airborne observations to initialize a 0-D model and compare adiabatic simulated outputs of cloud optical properties to those measured directly. The impact of entrainment is then identified by accounting for the differences between the simulated and observed cloud optical properties. To highlight these objectives, we state in the following text: "In this study, cloud droplet number concentration was not measured directly, and the closure study is conducted on cloud optical properties. RPAS measurements of cloud optical properties are more accurately reproduced by an ACPM simulation using a parameterization for entrainment compared to an adiabatic simulation."

In the manuscript current text : "The case studies in Cyprus (this study) and at Mace Head illustrate the significance of the entrainment processes in determining cloud optical properties in two different environments, and highlight notably that entrainment processes reduce the impact of cloud radiative forcing."

Updated text : "The case studies in Cyprus (this study) and at Mace Head illustrate the significance of the entrainment processes in determining cloud optical properties in two different environments. Entrainment mixing decreases the water content in the cloud relative to an adiabatic profile, which leads to a significant overestimation of cloud radiative forcing."

*Technical comments*
*Figure 10: Please rephrase the caption; "number of particles greater than 0.3 um" etc.*

Response: Captions have been changed, and now read :

In the manuscript current caption : "Figure 10. Vertical profiles of aerosol number concentration for a previous flight an hour earlier (Flight 65) greater 0.3 $\mu$m and greater 1 $\mu$m."

Updated text : "Figure 10. Vertical profiles of aerosol number concentration for number of particles greater than 0.3 $\mu$m and greater than 1 $\mu$m. Measurements of the aerosol number

concentration were conducted during a previous flight (Flight 65), which occurred an hour earlier than the flight considered in this study (Flight 67)."

*Figure 11: Please rephrase the caption; "mass concentration and chemical composition of aerosol during the flight ..." etc.*

In the manuscript current caption : "Figure 11. a) Concentration of aerosol chemical composition during the flight (Aerosol Chemical Speciation Monitor (ACSM) measurement). b) Normalized mass fraction of aerosol chemical composition during the flight."

Updated text : "Figure 11. a) Measurements from the Aerosol Chemical Speciation Monitor (ACSM) for the mass concentration of aerosols depending on their chemical composition. The time period covers the flight presently studied. b) Normalized aerosol mass concentration depending on aerosol chemical composition for the same period."

*Figure 13: Please explain the shaded areas briefly in the caption.*

In the manuscript current caption : "Figure 13. a) Simulated cloud droplet number function of the cloud height, for the adiabatic and the entrainment cases, with variation of updraft velocity (dw) and particle number (dN)."

Updated text : "Figure13. a) Simulated cloud droplet number function of the cloud height, for the adiabatic and the entrainment cases, with variation of updraft velocity ($dw$) and particle number ($dN$). Shaded areas are obtained from a variation of LWC of +/-0.05 g m$^{-3}$ in the calculation of cloud droplet number in the reference case, dark gray for the adiabatic case and light gray for the simulation with the entrainment parameterization. The intermediate gray corresponds to the intersection of the two cases."

*Figure 14: The x-axis label is pretty confusing, try to simplify.*

In the manuscript current figure : "Averaged normalized downwelling pyranometer and transmitted fraction of shortwave radiation"

Updated text: "Observed and simulated transmitted fraction of shortwave radiation"

*Page 3 lines 21-22 unclear sentence*

In the manuscript current text : "In Conant et al., (2004) and Meskhidze et al., (2005) closure studies on consistency between the variables $N_d$ , the impact of entrainment was observed; however, the data was screened and only on the case studies close to adiabatic values were studied."

Updated text : "In Conant et al. (2004) and Meskhidze et al. (2005), the impact of entrainment was observed; however, the data was screened and only the case studies approximating adiabatic

values were used to show aerosol-cloud closure of cloud droplet number concentrations near cloud base."

*Are the model code and data publicly available?*

Response: the data will be publicly available at the publication of the manuscript.

[revised manuscript text omitted]

---

## Author Response (AR2)

Comments to the Author:

One of the referees accepts your revised paper as it is. The other referee has a few comments which, because of some technical problems, are inserted below Please address these final, mostly minor issues.

**Editor**

Comments from the referee appear in italic, followed by responses from the authors and modification of the manuscript.

We thank the referee for his meticulous attention on this manuscript.

Yum et al. (2015) provides little support for inhomogeneous mixing. In the conclusion they state, "...the suggestion made here must remain speculative..." This is because, "The dominant feature was the positive relationship identified between V (cloud droplet mean volume) and LWC: 293 of 303 cases... This was a trait that would definitely be interpreted as homogeneous mixing." Or this could indicate no mixing. I am grateful that the authors directed my attention to this article because in our current analysis of two stratus cloud field experiments, we also find predominance of positive R between cloud droplet mean diameter (MD) and cloud droplet LWC in horizontal cloud penetrations; 94 of 97 in POST (the same 97 % as Yum et al.) and 19 of 19 in MASE. Near cloud top Yum et al. found one nonpositive VL relationship indicative of inhomogeneous mixing. Our POST analysis also showed less positive MD-L relationships at higher altitudes. These observations are then consistent with Brenguier et al. (2011), "...that entrainment has little impact on cloud microphysics except at cloud top..." Therefore, throughout most of stratus clouds, except possibly near cloud top, the inhomogeneous notion of consistent droplet spectral shape with LWC that is a foundational assumption of this manuscript does not apply.

Response: While the V and LWC relationship "would definitely be interpreted as homogeneous mixing" the remainder of the sentence in Yum et al. (2015) states "but estimation of the relevant scale parameters consistently suggested inhomogeneous mixing." Yum et al. (2015) also indicate that the relationship could be due to the preferential evaporation of smaller droplets upon descent inside the cloud (i.e., inhomogeneous mixing). In the reviewer's response, the reviewer has inserted why they think Yum et al 2015 said "the suggestion made here must remain speculative", yet, we also point out that the full statement is "... Nevertheless, the suggestion made here must remain speculative until more supporting evidence is accumulated." We interpret these remarks as suggesting the contribution of homogeneous and inhomogeneous mixing remains an unresolved issue.

In this manuscript, we do not have the measurements to explicitly claim one way or the other if our case is homogeneous or inhomogeneous. The inhomogeneous mixing assumption is made because, previous studies have indicated stratocumulus clouds are influenced by inhomogeneous mixing (e.g., Pawlowska et al., 2000; Burnet et al., 2006), and using recent advancements in technology (Jia et al., 2019). In addition, the inhomogeneous assumption yields results closer to our observations as it reduces the cloud optical thickness more than homogenous mixing. Therefore, the inhomogeneous assumption is utilized as a limit for the maximum reduction in cloud optical thickness due to cloud top entrainment. We have clarified this limiting factor in the text.

In the manuscript current text : "The reduction in number concentration due to entrainment is driven by the amount of evaporated water as we approximate the evaporation through inhomogeneous mixing (Jacobson et al., 1994). Figure 12b presents the profiles of LWC calculated from the ACPM in case of the adiabatic simulation and when the entrainment parameterization is considered."

Updated text : "The reduction in number concentration due to entrainment is driven by the amount of evaporated water as we approximate the evaporation through inhomogeneous mixing (Jacobson et al., 1994). In this study, the inhomogeneous assumption is utilized as a limit for the maximum reduction in cloud optical thickness due to cloud top entrainment. In addition, the inhomogeneous assumption yields results closer to our observations as it reduces the cloud optical thickness more than homogenous mixing. Figure 12b presents the profiles of LWC calculated from the ACPM in case of the adiabatic simulation and when the entrainment parameterization is considered."

Closure experiments of Conant et al. (2004), Peng et al. (2005), Fountoukis et al. (2007) and Sanchez et al. (2017) did use similar models. But closure in the former 3 was provided by in situ cloud microphysics measurements while Sanchez et al. (2017) provided satellite estimates of microphysics. The latter is weak closure, but I did not review that manuscript. The authors may be able to claim radiative closure but not cloud droplet concentration closure, especially when invoking inhomogeneous mixing. The unsupported assertion that inhomogeneous mixing is not "expected" to eliminate the CCN-CDNC relationship does not refute Twohy & Hudson (1995) that demonstrated this. Nor does it refute that good relationships between CCN concentrations and CDNC in subadiabatic cloud parcels is contrary to inhomogeneous mixing. This has been written in several papers (e.g., Hudson & Noble 2014; Hudson et al. 2018), and not challenged by any reviewers or readers.

Response: We thank the reviewer for his precision about using the term "CDNC-closure". In Sanchez et al., 2017, the closure statements were carefully worded to distinguish from the 'traditional' terms of aerosol-cloud closures based on available observations. As stated in the manuscript p7, line 13, the present work does not aim for a cloud droplet concentration closure; and

we have attempted to remove any ambiguity.

"In the present work, as no direct measurements of CDNC were available, the optical closure is addressed through the in-cloud fraction of transmitted shortwave radiation profile deduced from the ACPM and measured with the pyranometers."

Also, we are not refuting a CCN-CDNC relationship. While we cannot explicitly show a CCN-CDNC relationship when there is inhomogeneous entrainment, the literature contains evidence that inhomogeneous entrainment occurs in stratocumulus clouds (see previous comment). Again, we also suggested inhomogeneous mixing as results for the optical closure are closer to those of the observations (previous comment).

In the manuscript current text : "Section 3.3 Aerosol-CCN closure"

Updated text : "Aerosol-CCN comparison through the hygroscopicity parameter"

In the manuscript current text : "These values are in good agreement, and confirm a closure between aerosol physical and chemical properties and the CCN measurements."

Updated text : "These values are in good agreement, and confirm an acceptable coherency between aerosol physical and chemical properties and the CCN measurements."

In the manuscript current text : "Section 4 Aerosol-cloud closure study"

Updated text : "Optical-cloud closure study "

Hudson, J.G., and S. Noble, 2014: Low altitude summer/winter microphysics, dynamics and CCN spectra of northeastern Caribbean small cumuli; and comparisons with stratus. J. Geophys. Res., Atmos. 119, Issue 9, 16 May, 5445–5463, doi:10.1002/2013JD021442.

Hudson, J.G., S. Noble, and S. Tabor, 2018: CCN spectral shape and stratus cloud and drizzle microphysics. Journal of Geophysical Research: Atmospheres, 123, 9635-9651. http://doi.org/10.1029/2017JD027865

Twohy, C.H. and J.G. Hudson, 1995: Cloud condensation nuclei spectra within maritime cumulus cloud droplets. J. Appl. Meteorol., 34, 815-833.

Pawlowska, H. and Brenguier, J. (2000), Microphysical properties of stratocumulus clouds during ACE-2. Tellus B, 52: 868-887. doi:10.1034/j.1600-0889.2000.00076.x

Jia, Hailing, Ma, Xiaoyan, and Liu, Yangang. Exploring aerosol cloud interaction using VOCALS-REx aircraft measurements. United States: N. p., 2018. Web. doi:10.5194/acp-2018-667.

Burnet, F. and J. Brenguier, 2007: Observational Study of the Entrainment-Mixing Process in Warm Convective Clouds.J. Atmos. Sci.,64, 1995–2011, https://doi.org/10.1175/JAS3928.1

**Aerosol-Cloud Closure Study on Cloud Optical Properties using Remotely Piloted Aircraft Measurements during a BACCHUS Field Campaign in Cyprus**

Radiance Calmer1, Gregory C. Roberts1,2, Kevin J. Sanchez1,2,7, Jean Sciare3, Karine Sellegri4, David Picard4, Mihalis Vrekoussis3,5,6, and Michael Pikridas3

[revised manuscript text omitted]

ameter (nm) | particle number at the Hop-
pel minimum (cm -3 ) | number of particles at $0.24 \% SS \text{ (cm}^{-3})$ | diameter at 0.24 % SS (nm)     |
|-----------------------------------------------|-----------------------------------|----------------------------------------------------------------|-------------------------------------------------------|--------------------------------|
| 1234 (± 63.6)                                 | 100                               | 388
(min=366.8 max=408.3)                                   | 420                                                   | 91.15
(min=85.99 max=96.85) |

**Table 2.** Profile history from cloud base (1000 m a.s.l.) to the ceiling (2100 m a.s.l.) and down near cloud base again (950 m a.s.l.) duringthe flight

| Video time (min) | Altitude (m a.s.l.) | Observations                             |
|------------------|---------------------|------------------------------------------|
| 30:44            | 1072                | cloud base, start of the ascent profile  |
| 32:55            | 1295                | change in visibility, enter in the cloud |
| 36:17            | 1602                | cloud top                                |
| 39:40            | 1904                | below a convection cell                  |
| 41:00            | 2102                | maximum altitude of the profile          |
| 46:43            | 1730                | cloud cell, video-camera sees in cloud   |
| 51:42            | 1121                | first sight of ground                    |
| 53:11            | 996                 | cloud base, end of the descent profile   |

**Table 3.** Input values to calculate the sensitivity  $S_d$  used in Eq.10

|                           | adiabatic |       |       | entrainment |       |       |
|---------------------------|-----------|-------|-------|-------------|-------|-------|
|                           | ref       | dN    | dw    | ref         | dN    | dw    |
| Albedo*                   | 0.917     | 0.919 | 0.915 | 0.767       | 0.771 | 0.762 |
| CDNC* (cm -3 ) | 436.4     | 503.1 | 367.0 | 96.3        | 111.0 | 81    |
| LWC* (g $m^{-3}$ )        | 1.47      |       |       | 0.324       |       |       |

\*at cloud top

**Table 4.** Results of the sensitivity  $S_d$  of albedo for cloud droplet number concentration with variation of aerosol concentration (dN) and updraft (dw) given by Eq.10. The comparison of sensitivity calculation between adiabatic and entrainment cases at cloud top is  $S_{d_{ent}}$ =0.118

|          | adiabatic | entrainment |
|----------|-----------|-------------|
| $S_{dN}$ | 0.012     | 0.032       |
| $S_{dw}$ | 0.018     | 0.037       |